# PH domain-mediated autoinhibition and oncogenic activation of Akt

Hwan Bae[1,2,3†], Thibault Viennet[1,3†], Eunyoung Park[1,3], Nam Chu[1,2,3,4], Antonieta Salguero[1,2], Michael J Eck[1,3*], Haribabu Arthanari[1,3*], Philip A Cole[1,2*]

[1]Department of Biological Chemistry and Molecular Pharmacology, Harvard Medical School, Boston, United States; [2]Division of Genetics, Department of Medicine, Brigham and Women's Hospital,, Boston, United States; [3]Department of Cancer Biology, Dana-Farber Cancer Institute, Boston, United States; [4]Department of Cancer Biology and Genetics, The Ohio State University, Columbus, United States

**\*For correspondence:**
eck@crystal.harvard.edu (MJE);
hari_arthanari@hms.harvard.edu
(HA);
pacole@bwh.harvard.edu (PAC)

†These authors contributed
equally to this work

**Competing interest:** See page
23

**Reviewing Editor:** Amy H
Andreotti, Iowa State University,
United States

**Abstract** Akt is a Ser/Thr protein kinase that plays a central role in metabolism and cancer. Regulation of Akt's activity involves an autoinhibitory intramolecular interaction between its pleckstrin homology (PH) domain and its kinase domain that can be relieved by C-tail phosphorylation. PH domain mutant E17K Akt is a well-established oncogene. Previously, we reported that the conformation of autoinhibited Akt may be shifted by small molecule allosteric inhibitors limiting the mechanistic insights from existing X-ray structures that have relied on such compounds (Chu et al., 2020). Here, we discover unexpectedly that a single mutation R86A Akt exhibits intensified autoinhibitory features with enhanced PH domain-kinase domain affinity. Structural and biochemical analysis uncovers the importance of a key interaction network involving Arg86, Glu17, and Tyr18 that controls Akt conformation and activity. Our studies also shed light on the molecular basis for E17K Akt activation as an oncogenic driver.

## Editor's evaluation

Bae et al. provide fundamental new insight into molecular features associated with autoinhibited Akt; the Ser/Thr kinase that controls signaling pathways that drive cell survival, proliferation, and cancer. Akt structure/function studies have been ongoing for decades and the data presented in this work convincingly provide support for a new regulatory feature within Akt. As our understanding of Akt regulation is further refined by this and future work in the field, we can anticipate the picture of precisely how this and related kinases are controlled at the molecular level will continue to emerge.

## Introduction

The Ser/Thr protein kinase Akt1 (termed Akt in the present work) and its paralogs Akt2 and Akt3 serve as key effectors in the PI3-kinase/Akt signaling pathway to regulate cell survival, proliferation, and metabolism by phosphorylating a variety of protein substrates (**Manning and Toker, 2017**). Akt is a 480 amino acid protein comprised of an N-terminal pleckstrin homology (PH) domain, a central kinase domain, and a regulatory C-tail. Akt's PH domain shows high affinity and selectivity for the phospholipid, phosphatidylinositol 3,4,5-triphosphate (PIP3) using its positively charged basic residues including Lys14, Arg23, Arg25, and perhaps to a limited extent Arg86 (**Milburn et al., 2003**). It has been suggested that PIP3 binding to Akt's PH domain principally drives cell signaling by directly and allosterically turning on Akt's kinase activity (**Ebner et al., 2017**; **Lučić et al., 2018**; **Truebestein et al., 2021**), but our work and that of others do not support such a model (**Alessi et al., 1996**; **Balasuriya et al., 2018**; **Chu et al., 2018**; **Chu et al., 2020**; **Cole et al., 2019**; **Zhang et al., 2006**).

Recruitment of Akt to the plasma membrane by PIP3 leads to its phosphorylation on two key sites, Thr308 in Akt's activation loop by PDK1 and Ser473 in the regulatory C-tail by mTORC2 (*Alessi et al., 1997*; *Sarbassov et al., 2005*). The Thr308 and Ser473 phosphorylation events (pT308 and pS473) stimulate Akt kinase activity by relieving autoinhibition (*Cole et al., 2019*; *Manning and Toker, 2017*). Termination of Akt signaling is known to be mediated by the protein phosphatases, PHLPP for p473 hydrolysis, and PP2A for pT308 hydrolysis (*Gao et al., 2005*; *Kuo et al., 2008*; *Seshacharyulu et al., 2013*).

In the current structural model, even for an isolated Akt kinase domain, phosphorylation of Thr308 is critical to reach its active catalytic conformation whereas pS473 effects are only apparent in the full-length enzyme and relieve inhibitory intramolecular interactions between the kinase and PH domain (*Alessi et al., 1996*; *Calleja et al., 2009*; *Chu et al., 2018*; *Chu et al., 2020*; *Cole et al., 2019*; *Yang et al., 2002a*; *Yang et al., 2002b*). The pS473 has been shown to stimulate Akt by phosphate-mediated concurrent interactions to the side chains of Gln218 in the N-lobe of the kinase domain and Arg144 in the PH-kinase linker region (*Chu et al., 2018*). Recent work suggests that non-canonical C-terminal post-translational modifications such as dual phosphorylation of Ser477 and Thr479 (pS477/pT479) in place of Ser473 may also relieve Akt autoinhibition through distinct but incompletely understood molecular mechanisms (*Chu et al., 2018*; *Liu et al., 2014*; *Salguero et al., 2022*).

PIP3 production is governed by the action of receptor tyrosine kinases like the insulin receptor which in the presence of agonists stimulate PI3-kinase that catalyzes the conversion of phosphatidylinositol 4,5-bisphosphate (PIP2) to PIP3 (*Hoxhaj and Manning, 2020*; *Yang et al., 2019*). PIP3 hydrolysis is tightly controlled by the lipid phosphatase PTEN (*Chalhoub and Baker, 2009*; *Dempsey et al., 2021*). Akt is frequently hyperactivated in human cancers (*Hua et al., 2021*). The hyperactivation of Akt in cancer has led to intensive therapeutic discovery efforts to identify Akt ATP site and allosteric inhibitors although none has yet progressed to FDA approval (*Janku et al., 2018*; *Luo et al., 2003*). Hyperactivation of Akt pathway in cancer occurs most commonly through gain-of-function mutations in PI3-kinase or loss-of-function mutations in PTEN phosphatase (*Lee et al., 2018*; *Shariati and Meric-Bernstam, 2019*; *Yang et al., 2019*), but oncogenic Akt mutations are also observed in 2–5% of many solid tumors. The Akt E17K variant represents approximately 90% of all mutant Akt forms in human cancer (*Shariati and Meric-Bernstam, 2019*). The molecular basis of neoplasia associated with Akt E17K mutation is reported to derive from Akt E17K's enhanced phospholipid affinity and membrane localization relative to WT Akt, rather than through direct kinase stimulation (*Carpten et al., 2007*; *Landgraf et al., 2008*).

Recent efforts to interrogate Akt's molecular mechanisms have relied on the technique of expressed protein ligation (EPL) to generate site-specific C-tail modified forms of Akt (*Chu et al., 2018*; *Chu et al., 2020*; *Salguero et al., 2022*). This method involves the chemoselective ligation of a recombinant protein thioester with an N-Cys containing C-tail peptide (*Muir et al., 1998*). In addition, EPL has also been used to generate PH-domain isotopically labeled Akt for NMR studies using a sequential three-piece ligation strategy (*Chu et al., 2020*).

Several prior studies have attempted to elucidate the structural basis of PH domain-mediated autoinhibition of Akt, and two types of high-resolution structures of near full-length Akt in distinct 'closed' conformations have been obtained by X-ray crystallography (*Ashwell et al., 2012*; *Quambusch et al., 2019*; *Truebestein et al., 2021*; *Wu et al., 2010*). These X-ray structures have been obtained in complex with small molecule allosteric inhibitors or PH-kinase linker-bound nanobodies, showing tight interaction between the PH and kinase domains. However, each structure showed different key interacting residues at the PH-kinase domain interface depending on which ligand the structure includes. It has not yet been possible to obtain high-resolution X-ray crystal structures of full-length autoinhibited Akt in the absence of such ligands, although NMR studies have suggested that allosteric inhibitors may perturb the baseline Akt autoinhibited state (*Chu et al., 2020*).

In this study, we have explored the PH domain-mediated autoinhibition of Akt using an array of biochemical and structural approaches. Key to this work was the unexpected identification of an Akt PH domain mutant, R86A, which we posit sheds light not only on PH domain-mediated Akt autoinhibition but also on how the oncogenic mutant E17K activates Akt.

## Results

### Analysis of R86A Akt and autoinhibition

Non-classical Akt activation by dual C-terminal phosphorylation of Ser477 and Thr479 appears to employ a distinct molecular mechanism compared with classical Ser473 phosphorylation (*Figure 1A*; *Cole et al., 2019*). In contrast to pS473 Akt, pS477/pT479 Akt displays a weakened affinity for PIP3, hinting that perhaps the pS477 and/or pT479 groups directly contact the PH domain as part of their Akt kinase activation effects (*Chu et al., 2018*). Inspection of the known crystal structure of the Akt PH domain suggested three basic surfaces on the PH domain that could potentially interact with C-terminal phosphorylation sites, Lys30/Arg48, Arg15/Lys20/Arg67, and Arg86 (*Figure 1B*; *Milburn et al., 2003*). To investigate the possibility that these basic surfaces might be key for regulation, we prepared a series of semisynthetic mutant Akt forms containing pS477/pT479 (*Figure 1—figure supplement 1*, Akt constructs named with A1–A4). To generate these Akt forms, we employed a sequential, three-piece ligation strategy akin to a previous approach (*Figure 1C and D*). Thr308 phosphorylation of these proteins was accomplished using in vitro PDK1 treatment. We then examined the catalytic activity of these mutant PH, pS477/pT479 Akt forms using GSK3 peptide as a substrate with radioactive ATP assays. These assays revealed that the K30A/R48A (A3, V/E 15 min$^{-1}$) and R15A/K20A/R67A Akt (A4, V/E 11 min$^{-1}$) forms showed similar kinase rates to WT Akt (A1, V/E 12 min$^{-1}$) prepared under the same conditions (*Figure 2A*). In contrast, R86A Akt (A2, V/E 3.4 min$^{-1}$) displayed an approximately fourfold reduction in enzymatic activity. This suggested that the Arg86 side chain would be a candidate for electrostatic interaction with pS477 and/or pT479 to induce Akt activation. To explore this further, we generated R86A pS473 and non-phosphorylated C-tail Akt and unexpectedly found that Arg86 mutation suppressed the catalytic activity of pS473. In this set of measurements R86A pS473 (A11, V/E 2.7 min$^{-1}$) was about fourfold lower than WT pS473 (A10, V/E 9.1 min$^{-1}$) (*Figure 2B*). These results suggest that R86A mutation can generally augment the autoinhibitory interaction between the PH and kinase domains.

To further characterize the R86A-mediated effects, we assessed the intermolecular affinity of the isolated R86A PH domain (aa 1–121) for the Cy5-labeled kinase domain (S122C, aa 122–480, pT308, non-C-tail phosphorylated) using microscale thermophoresis (MST). As shown in *Figure 2C*, the R86A PH domain displayed an approximately threefold greater binding affinity (K$_D$ 37 µM) toward the kinase domain than the WT PH domain (K$_D$ 107 µM), supporting the idea that R86A mutation stabilizes the autoinhibited Akt state by promoting the PH-kinase interdomain interaction. We also attempted to measure the affinity of the R86A PH domain to the kinase domain in the presence of the Akt allosteric inhibitor MK2206 using MST but were unable to see detectable binding (K$_D$ > 250 µM). This was an unexpected result since MK2206 has been shown to enhance the affinity of WT PH domain to the kinase domain, previously (*Chu et al., 2020*). We thus interpret the antagonistic behavior between R86A and MK2206 on the PH-kinase domain interaction to suggest that MK2206 induces a gluing of the PH domain and kinase domain that is distinct from the effects of R86A.

### Structural analysis of the R86A PH domain

To gain insight into how mutation of Arg86 impacted the protein domain's three-dimensional structure, we initially pursued an NMR approach with an $^{15}$N-labeled PH domain. Remarkably, the 2D $^1$H-$^{15}$N HSQC NMR analysis of the R86A PH domain displayed significant chemical shift perturbations (CSPs) relative to the WT PH domain (*Figure 3A*). We observed that the R86A PH domain is much more stable compared to the WT. Differential scanning fluorimetry revealed that R86A has increased melting temperature (51±1°C) compared to WT PH domain (42±1°C, *Figure 3—figure supplement 1*). Specifically the NMR samples of the R86A PH domain at concentrations of 50–100 µM were stable for greater than 3 weeks when maintained at room temperature whereas the WT PH domain in the same conditions undergoes substantial aggregation in less than 24 hours. Due to this increased stability, we were able to assign 81% of the backbone resonances of the R86A PH domain compared to 66% of the WT PH domain. The CSPs between the NMR spectrum of the isolated R86A PH domain and the WT PH domain (*Figure 3A–C*) indicate chemical shift changes centered around the mutation site, spreading to the amino acids in the structural neighborhood. In this case a basic amino acid with a long side chain, Arg, is replaced with an Ala, so some of these CSPs are expected. However, we also noticed CPSs in relative distal portions of the protein, away from the mutated site, which indicates that

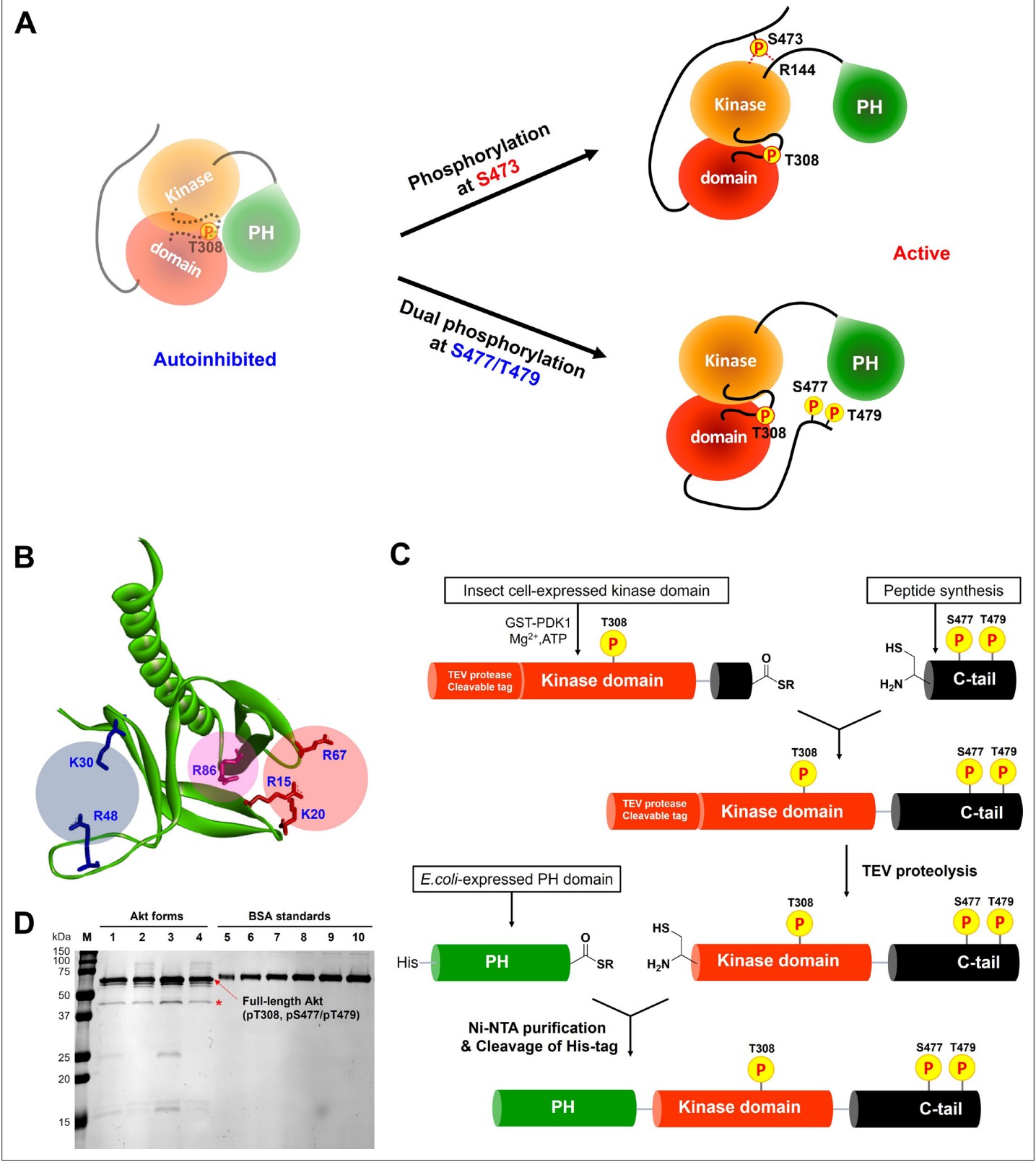

**Figure 1.** Distinct potential Akt regulation mechanisms associated with pS473 and pS477/pT479, and a semisynthetic strategy to generate site specifically modified Akt forms. (**A**) Two different proposed Akt activation mechanisms by C-tail phosphorylation events at Ser473 and Ser477/Thr479. (**B**) Inspection of the crystal structure of the Akt PH domain (PDB:1UNP) identified three potential pSer477/pThr479 basic binding surfaces in the PH domain highlighted in colored circles. (**C**) Three-piece expressed protein ligation strategy to make full-length Akt forms with site-specific

*Figure 1 continued on next page*

*Figure 1 continued*

and stoichiometric phosphorylation at Thr308, Ser477, and Thr479 as well as different alanine mutations on their PH domains. (**D**) The purity and concentration of each Akt form determined using SDS-PAGE followed by Coomassie staining. Lane 1: WT (**A1**), lane 2: R86A (**A2**), lane 3: K30A R48A (**A3**), lane 4: R15A K20A R67A (**A4**), lanes 5–10: BSA standards 0.1, 0.2, 0.3, 0.4, 0.5, 0.6 µg, M: protein markers (kDa). * Mark indicates PreScission Protease added to cleave the N-terminal tags, which does not appear to alter the Akt catalytic activities.

The online version of this article includes the following source data and figure supplement(s) for figure 1:

**Source data 1.** Raw gel image showing the purity and concentration of each Akt mutant.

**Figure supplement 1.** List of semisynthetic Akt constructs.

there could be additional changes in conformation and/or dynamics between the R86A PH domain and the WT PH domain.

We also determined the X-ray crystal structure of the R86A PH domain at 1.39 Å resolution (PDB: 7MYX, *Table 1*, *Figure 3F*, and *Figure 3—figure supplement 2*). Despite the NMR HSQC analysis suggesting the possibility of structural perturbations induced by R86A mutation, the X-ray structure of the R86A PH domain revealed very high concordance between the WT and R86A PH domain structures, with an overall RMSD of 0.198 for main chain atoms (*Figure 3D and E*). The structure of the R86A mutant was determined in the same crystal lattice as the WT protein, and their main chain conformations are essentially identical. In fact, apart from the mutated Arg86 residue, only the side chains of Glu17 and Tyr18 showed significant structural changes in the R86A X-ray structure (*Figure 3E*). In the WT PH domain structure (PDB: 1UNP), the carboxylate of Glu17 forms electrostatic interactions with the guanidinium of Arg86 and the side chain of Lys14 (*Figure 3D*; *Milburn et al., 2003*). By contrast, in the R86A PH domain, the Glu17 side chain rotates away from Ala86 and the side chain of Lys14, and this rearrangement appears to induce an alternate side chain rotamer for Tyr18 (*Figure 3E*).

Though the X-ray structures of the WT and R86A PH domains are very similar, the NMR CSPs at locations distal to the mutation site could be explained by changes in intrinsic dynamics. One observation from the WT PH domain $^1$H-$^{15}$N HSQC spectrum (*Figure 3A*) supports this hypothesis. A few resonances present a secondary, weaker, signal (*Figure 3—figure supplement 3A*). These weak

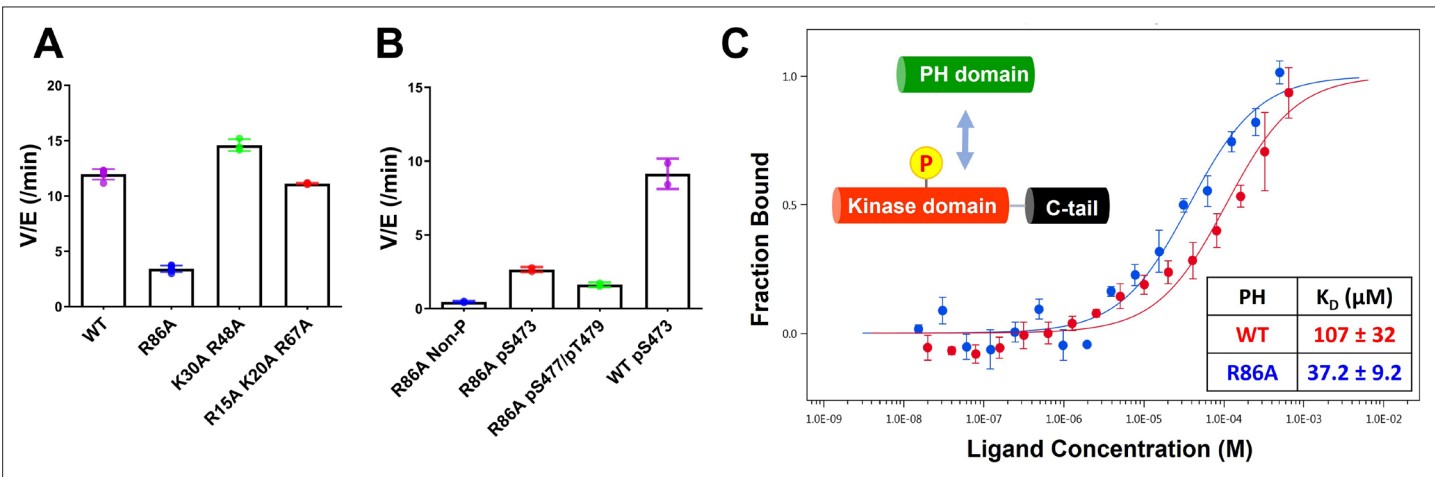

**Figure 2.** Kinase activities of Akt PH domain basic residue mutations and analysis of R86A PH domain intermolecular binding to the Akt kinase domain. (**A**) Enzymatic activities of full-length Akt mutants possessing pT308 and pS477/pT479 (A1–A4) prepared using a three-piece expressed protein ligation (EPL) strategy. (**B**) Enzymatic activities of full-length R86A Akt mutant forms with differentially phosphorylated C-tails (A5: pS477/pT479, A7: Non-P, A11: pS473) relative to WT Akt containing pS473 (A10) as a control. These Akt forms were prepared using a two-piece EPL strategy. These kinase assays were performed in buffer containing 250 µM ATP and 20 µM GSK3 peptide as substrates (n≥3, SD shown). (**C**) MST (microscale thermophoresis) binding experiments using the N-terminally Cy5-labeled kinase domain with pT308 as a target protein and the isolated PH domain (WT or R86A) as a ligand. WT: red, R86A: blue (n=3, SEM shown).

The online version of this article includes the following source data for figure 2:

**Source data 1.** Kinase activity assays with full-length Akt mutants having pT308 and pS477/pT479.

**Source data 2.** Kinase activity assays with full-length R86A Akt mutant forms with differentially phosphorylated C-tails.

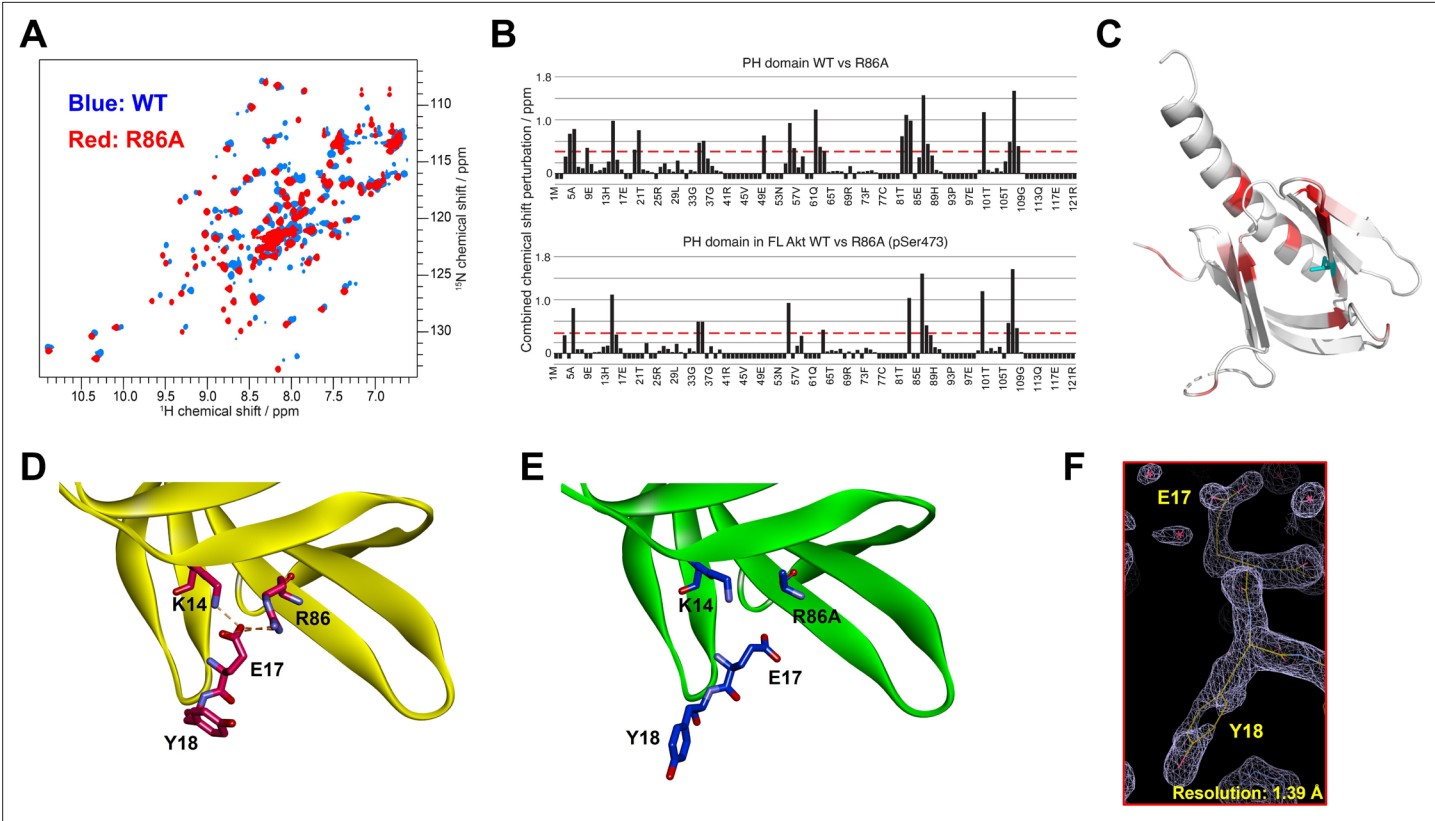

**Figure 3.** Structural studies on the R86A PH domain using NMR and X-ray crystallography. (**A**) $^1$H-$^{15}$N HSQC NMR spectra of $^{15}$N-labeled isolated PH domain. WT: blue, R86A: red. (**B**) Quantified chemical shift perturbations (CSPs) by R86A. Combined CSPs are plotted along the PH domain primary sequence for R86A isolated PH domain (top) and R86A PH domain in full-length Akt (bottom) referenced to the corresponding WT constructs. Red dash lines correspond to the standard deviation to the mean, excluding outliers (higher than ×3 SDM). Negative bars (–0.1) indicate nonassigned residues. (**C**) CSPs between WT and R86A PH domains corresponding to spectra in (**A**) are represented on a white (not significant) to red (maximum CSP) scale on the X-ray structure of R86A PH domain. Ala86 is shown in cyan. Most CSPs form a disk around the mutation site, but longer-range effects are seen as well. (**D**) X-ray crystal structure of the WT PH domain (PDB: 1UNP) or (**E**) the R86A PH domain (PDB: 7MYX). (**F**) The 2Fo–Fc omit map contoured at 1 sigma around E17 and Y18 residues of the isolated R86A PH domain.

The online version of this article includes the following source data and figure supplement(s) for figure 3:

**Source data 1.** Quantification of chemical shift perturbations by R86A in the isolated PH domain or the PH domain in full-length Akt.

**Figure supplement 1.** Thermal stability of PH domains.

**Figure supplement 2.** The structure of the R86A PH domain with b-factor values.

**Figure supplement 3.** R86A removes WT PH domain dynamics.

**Figure supplement 3—source data 1.** NMR relaxation studies on the R86A PH domain.

**Figure supplement 4.** Rigidity of PH domain R86A mutant.

**Figure supplement 5.** Modes of interaction of WT and R86A PH domains to Akt kinase domain.

signals could be due to a minor conformation of the same residue or could correspond to an unassigned residue. We investigated these weak resonances as function of temperature and observed that they collapse into a single signal at lower temperatures, supporting the case of multiple exchanging conformations. Furthermore, the residues that display multiple conformations cluster around R86 in a disk-like arrangement reminiscent of the CSPs between WT and R86A (*Figure 3—figure supplement 3B*). Taken together with its lower stability and melting temperature, this collectively suggests the possibility of multiple exchanging conformations in WT PH domain. However, the limited stability of the sample impedes further investigation by NMR, thus we cannot fully confirm the presence of multiple conformations.

It should be noted that the $^1$H-$^{15}$N HSQC spectrum of the R86A PH domain does not contain these weak, secondary, signals (*Figure 3—figure supplement 3D*), which suggest that R86A mutation

**Table 1.** Data collection and refinement statistics.

| | Akt R86A PH domain, PDB ID: 7MYX |
|---|---|
| **Data collection\*** | |
| Space group | C 1 2 1 |
| Cell dimensions | |
| a, b, c (Å) | 84.8, 33.6, 42.4 |
| α, β, γ (°) | 90.0, 119.3, 90.0 |
| Resolution (Å) | 36.96–1.39 (1.44–1.39) |
| No. reflections | 68960 (6254) |
| No. unique reflections | 20816 (1987) |
| $R_{meas}$ | 0.0487 (0.134) |
| $R_{merge}$ | 0.0409 (0.111) |
| $R_{pim}$ | 0.0261 (0.074) |
| I/σ (I) | 32.0 (9.60) |
| $CC_{1/2}$ (%) | 99.6 (98.2) |
| Completeness (%) | 97.2 (95.1) |
| Redundancy | 3.3 (3.1) |
| **Refinement** | |
| Resolution (Å) | 36.96–1.39 |
| No. reflections | 20,568 |
| $R_{work}$/$R_{free}$ | 0.18/0.20 (0.20/0.23) |
| No. atoms | |
| Protein | 958 |
| Ligand | – |
| Water | 111 |
| B factors | |
| Protein | 16.7 |
| Ligand | – |
| Water | 27.2 |
| R.m.s. deviations | |
| Bond lengths (Å) | 0.006 |
| Bond angles (°) | 0.80 |
| Ramachandran | |
| Preferred (%) | 98.17 |
| Allowed (%) | 1.83 |
| Outliers (%) | 0 |

A single crystal was used to collect data. \*Values in parentheses are for the highest-resolution shell.

removes some of the WT dynamics. The improved stability of R86A PH domain allowed us to conduct relaxation-dispersion experiments that further confirmed the absence of dynamics on the μs-ms timescale (*Figure 3—figure supplement 3E*). We went on and measured the $R_1$, $R_2$, $Eta_{xy}$ relaxation rates and heteronuclear Overhauser effect (hetNOE) for the R86A mutant (*Figure 3—figure supplement 4*). These relaxation rates show high rigidity (most hetNOE values > 0.65 and calculated correlation time of 8.7 ns, expected for a globular, rigid protein of 14 kDa). Together with the observed high stability and melting temperature of R86A, this proves the absence of multiple exchanging conformations in the R86A PH domain.

Thus there is a strong possibility that the WT PH domain, in contrast to the R86A mutant, is significantly more dynamic and that this could be the reason for the significant CSPs. These results may suggest that the PH domain conformation of the isolated R86A PH domain is preorganized before binding to the kinase domain to effectively autoinhibit Akt.

We then went on to generate $^{15}$N-segmentally labeled (aa 1–121) full-length R86A Akt using our semisynthetic strategy. Here, $^{15}$N-labeled R86A PH domain was ligated to the Akt kinase domain harboring a pS473 in the C-terminal tail. We compared the CSPs on the R86A PH domain due to the interaction of the kinase domain and C-terminal tail with the CSPs on the WT PH domain under the same conditions (*Figure 3—figure supplement 5A,B*). The average CSPs on the PH domain in the context of FL Akt are higher in the case of the R86A mutant, in line with the higher binding affinity. In both cases, we observe similar patterns of CSPs and peak broadening indicative of the interaction of the PH domain with the rest of the protein. Specifically, aa 14–22 show high CSPs or peak broadening in both cases and this region is expected to be in the core of the binding interface to the kinase domain and aa 81–90 similarly show indications of binding, as expected from our previous model (*Chu et al., 2020*). Though the comparative CSP patterns of the R86A and WT PH domains when anchored to the rest of Akt look similar, there are subtle differences around residues 67–73 (*Figure 3—figure supplement 5C*), which suggests that the R86A mutation might reorganize the interaction surface and/or affinity. It should be noted that this section of the PH domain is not thought to be part of the interaction interface, based on the structure of the allosterically inhibited Akt (PDB: 3O96). A

complete comparative analysis of the CSPs is limited by the fact that we have only 66% backbone resonance assignment for the WT PH domain. Moreover, we observed that the residues that experience CSPs as a result of the R86A mutation in the isolated PH domain still experience CSPs of similar magnitudes due to the R86A mutation when the PH domain is ligated to the rest of Akt (*Figure 3B*). Notably, these R86A-driven CSPs have a higher magnitude compared to the CSPs due to the interaction with the kinase domain. This supports our hypothesis that the R86A PH domain is preorganized in a rigid, stable, conformation suited for tight inhibitory interaction with the kinase domain.

## Effects of Glu17 and Tyr18 mutations on Akt PH-kinase domain interaction and catalytic activity

The structural analysis of the R86A PH domain led to the hypothesis that rotamer changes of Glu17 and Tyr18 might play important roles in mediating the enhanced autoinhibition conferred by R86A. Indeed, as discussed, the E17K Akt mutant is known to be oncogenic. To investigate this, we measured the intermolecular affinities of the Akt kinase domain for the E17K/R86A PH domain ($K_D$ 101 μM) and the Y18A/R86A PH domain ($K_D$ 210 μM), approximately threefold and approximately sixfold weaker than that of the R86A PH domain ($K_D$ 37 μM), respectively (*Figure 4A*). These results are consistent with the idea that Glu17 and Tyr18 of the PH domain play key roles in engaging the Akt kinase domain. To evaluate whether the hydroxyl or the phenyl ring of Tyr18 is most critical for binding to the kinase domain, we also measured the kinase domain's binding affinity to the Y18F/R86A PH and the E17K/Y18F/R86A PH domains. Interestingly, the affinity of Y18F/R86A PH ($K_D$ 45 μM) was very similar to that of the R86A PH domain ($K_D$ 37 μM) and the E17K/Y18F/R86A PH ($K_D$ 21 μM) was approximately twofold tighter (*Figure 4—figure supplement 1A,B* ), indicating that the aromatic ring rather than the hydroxyl of Tyr18 is most critical for mediating kinase interaction. Moreover, the enhanced affinity of the E17K/Y18F/R86A PH for the kinase domain is consistent with the possibility that the Tyr18 hydroxyl forms a hydrogen bond with the Lys17 side chain as observed in an X-ray structure of the E17K PH domain (*Figure 4—figure supplement 1C*; *Carpten et al., 2007*).

We next prepared full-length E17K, Y18A, and R86A Akt forms (*Figure 1—figure supplement 1*) containing pT308 with or without pS473 by EPL using a two-piece ligation strategy and confirmed the site-specific phosphorylation status of each Akt form using western blot analysis (*Figure 4—figure supplements 2 and 3*). We went on to measure the steady-state kinetic parameters of each of these Akt forms, determining the ATP $K_M$ values, as prior work has demonstrated that ATP $K_M$ rather than peptide substrate $K_M$ is most sensitive to PH domain-mediated autoinhibition (*Chu et al., 2018*; *Chu et al., 2020*). The catalytic efficiencies of non-C-tail phosphorylated Akt forms of E17K (A8, $k_{cat}/K_M$ 0.012 min$^{-1}$ μM$^{-1}$) and Y18A (A9, $k_{cat}/K_M$ 0.088 min$^{-1}$ μM$^{-1}$) were 2.5-fold and 18-fold elevated with respect to WT (A6, $k_{cat}/K_M$ 0.0048 min$^{-1}$ μM$^{-1}$) whereas the catalytic efficiency of the R86A mutant (A7, $k_{cat}/K_M$ 0.0024 min$^{-1}$ μM$^{-1}$) was reduced by twofold (*Figure 4B*). These results illustrate the importance of Tyr18 in driving Akt autoinhibition in the absence of pS473. The catalytic efficiencies of pS473 Akt forms of E17K (A12, $k_{cat}/K_M$ 0.12 min$^{-1}$ μM$^{-1}$) and Y18A (A13, $k_{cat}/K_M$ 1.1 min$^{-1}$ μM$^{-1}$) were 1.7-fold and 15-fold increased compared to WT (A10, $k_{cat}/K_M$ 0.071 min$^{-1}$ μM$^{-1}$) which was threefold greater than R86A (A11, $k_{cat}/K_M$ 0.024 min$^{-1}$ μM$^{-1}$). These results are consistent with the dominance of Tyr18 in mediating Akt autoinhibition, even when partially relieved by pS473 (*Figure 4C*).

## Effects of mutations on Akt binding to phospholipids and sensitivity to phosphatases

To increase our understanding of the biochemical properties of the E17K, Y18A, and R86A PH domains, we measured their affinities for fluorescein-labeled soluble (C8) PIP3 using fluorescence anisotropy. It was found that PIP3 affinity for E17K ($K_D$ 0.22 μM) was essentially identical to WT ($K_D$ 0.26 μM) and much weaker for Y18A ($K_D$ 4.5 μM) and R86A ($K_D$ 49 μM; *Figure 5A*). Interestingly, the double mutant E17K/Y18A PH led to restore the affinity for PIP3 ($K_D$ 0.30 μM), similar to WT (*Figure 5A*). These results suggest that the loss of PIP3 binding by Tyr18 mutation results from a PH domain conformational change rather than a direct interaction between Tyr18's side chain and PIP3. We also measured the binding affinity of WT and E17K PH domain interactions with soluble fluorescein-labeled PIP2 (C6). PIP2 binding to the E17K PH domain ($K_D$ 5.6 μM) was approximately fivefold tighter relative to its engagement of the WT PH domain ($K_D$ 31 μM; *Figure 5—figure supplement 1A*). Although the E17K PH domain binds PIP2 about 25-fold more weakly than PIP3, given the greater membrane abundance

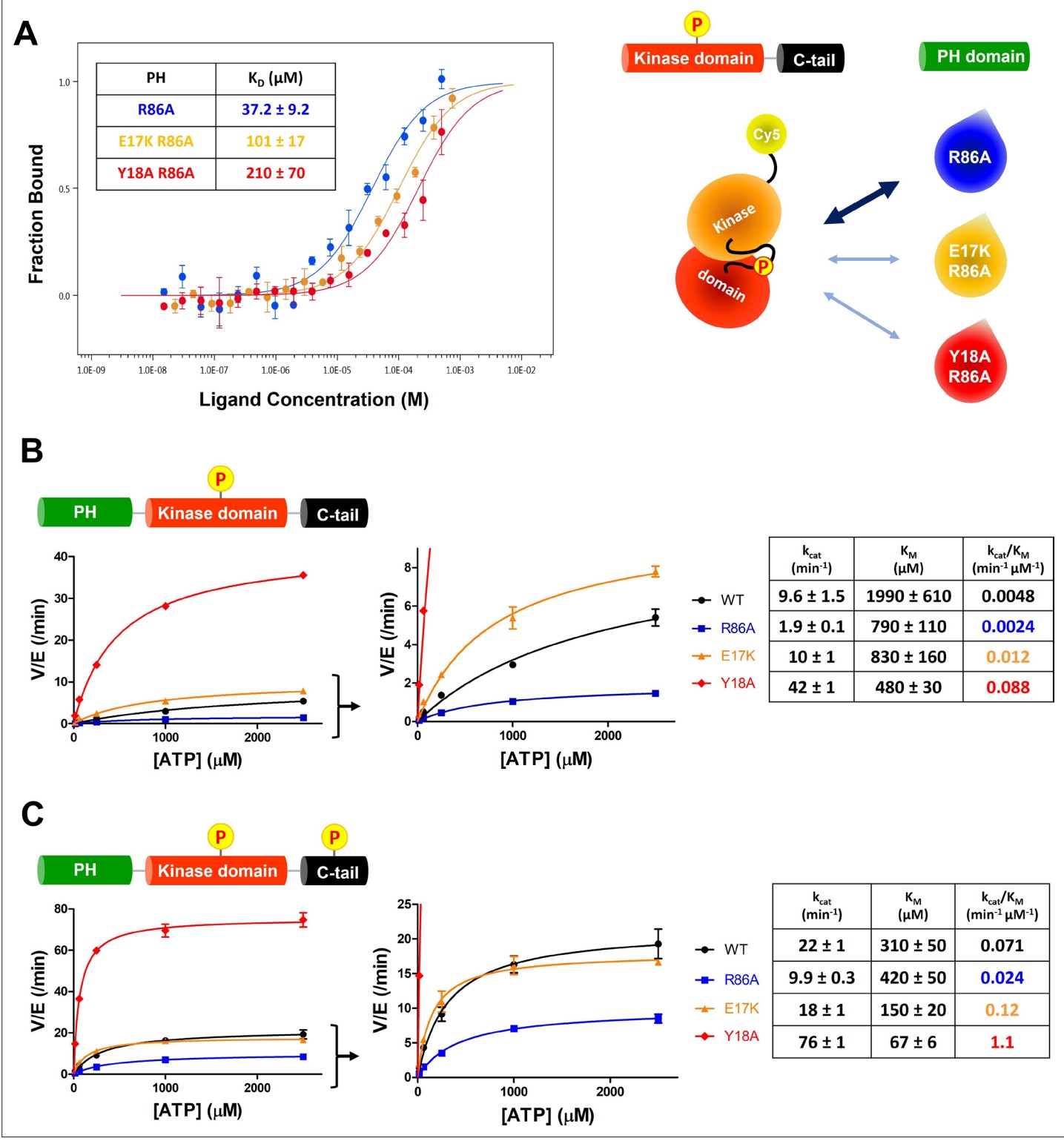

**Figure 4.** E17K and Y18A mutations promote the catalytic activity of Akt and disrupt the interdomain interactions between the PH and kinase domains. (**A**) MST (microscale thermophoresis) binding experiments using N-terminally Cy5-labeled kinase domain with pT308 as a target protein and isolated PH domain as a ligand. R86A: blue, E17K/R86A: yellow, Y18A/R86A: red (n=3, SEM shown). (**B**) Steady-state kinetic plots of full-length Akt mutants with pT308 (**B**) in the absence or (**C**) presence of a pS473 C-tail modification (A6–A13). Kinase assays were performed with each Akt mutant in buffer containing 20 µM GSK3 peptide substrate and varying amounts of ATP (0–2.5 mM). Kinetic parameters of each Akt mutant were determined from V/[E] versus [ATP] plots (n=2, SEM shown).

*Figure 4 continued on next page*

*Figure 4 continued*

The online version of this article includes the following source data and figure supplement(s) for figure 4:

**Source data 1.** Kinase activity assays with full-length Akt mutants (WT, R86A, E17K, Y18A) having pT308 and non-phosphorylated C-tail.

**Source data 2.** Kinase activity assays with full-length Akt mutants (WT, R86A, E17K, Y18A) having pT308 and pS473.

**Figure supplement 1.** The role of the hydroxyl group of Tyr18 in the PH-kinase domain interaction.

**Figure supplement 2.** Semisynthesis of Akt mutants containing pT308 in the absence or presence of pS473 using a two-piece ligation strategy.

**Figure supplement 2—source data 1.** Raw western blot images showing the phosphorylation levels at Ser473 and Thr308 of each Akt construct.

**Figure supplement 3.** Validation of full-length Akt mutant proteins.

of PIP2 versus PIP3 (*Wang and Richards, 2012*), this increased affinity may lead to enhanced plasma membrane localization of E17K in the absence of growth factors.

We next investigated the susceptibility of the various mutant forms of full-length Akt containing pT308 or a combination of pT308/pS473 Akt to dephosphorylation by PP2A and separately by the generic enzyme alkaline phosphatase. The dephosphorylation kinetics were monitored by densitometry of western blots with phospho-site antibodies. For pT308 dephosphorylation by PP2A, E17K (A8, $T_{1/2}$ 60 min$^{-1}$) and Y18A (A9, $T_{1/2}$ 67 min$^{-1}$) Akt forms displayed about 2.5-fold slower dephosphorylation, respectively, relative to WT Akt (A6, $T_{1/2}$ 24 min$^{-1}$) (*Figure 5B*). In contrast, PP2A hydrolysis of pT308 in R86A Akt (A7, $T_{1/2}$ 19 min$^{-1}$) was slightly (but not significantly) faster than WT Akt (*Figure 5B*). Alkaline phosphatase treatment of the pT308/pS473 Akt forms (A10–A13) showed no effect on pT308 but readily cleaved pS473 with no discernible difference in kinetics between the WT and mutant Akt forms (*Figure 5—figure supplement 1B*). Taken together, these results indicate that pS473 is similarly exposed among the various PH mutant forms whereas pT308 is differentially protected by the intramolecular interactions.

## Phe309 and Akt autoinhibition

As described above, we have obtained strong evidence that Tyr18 is a critical player in the autoinhibitory interaction between PH and kinase domains. Inspection of prior X-ray crystal structures of near full-length Akt in complex with allosteric inhibitors shows that Tyr18 is in close proximity to the catalytic core of the Akt kinase domain (*Ashwell et al., 2012*; *Quambusch et al., 2019*; *Wu et al., 2010*), but we were not able to identify an obvious hydrophobic interaction between Tyr18 and the kinase domain that could govern the PH domain-mediated autoinhibition. A more recent crystal structure places Tyr18 near a hydrophobic pocket comprised of Leu316, Val320, and Leu321 (*Truebestein et al., 2021*), and L321A Akt appears to be activated, while L316A and V320A are not (*Parikh et al., 2012*). However, this crystal structure analyzes a chimeric Akt form with a shortened and mutated PH-kinase linker in a complex with a nanobody (PDB:7APJ) and lacks electron density around the activation loop and the N-lobe of the kinase domain, casting uncertainty about its relevance to the nanobody-free natural enzyme. To gain an additional perspective, we examined the AlphaFold-predicted full-length Akt structure (*Jumper et al., 2021*; *Varadi et al., 2022*). The structural prediction of AlphaFold showed an unexpected close contact between Tyr18 in the PH domain and Phe309 on the activation loop of the Akt kinase domain (*Figure 6A* and *Figure 6—figure supplement 1*), which was not shown in the reported near full-length Akt structures in complex with small molecule allosteric inhibitors or a linker-bound nanobody having insufficient electron density to visualize Phe309 on the activation loop (*Ashwell et al., 2012*; *Quambusch et al., 2019*; *Truebestein et al., 2021*; *Wu et al., 2010*). Tyr18 and Phe309 are highly conserved residues phylogenetically across Akt forms. Moreover, when this AlphaFold structure is superimposed with the crystal structure of the active kinase domain complexed with the GSK3β peptide substrate (PDB: 6NPZ; *Chu et al., 2018*), the Tyr18-Phe309 interaction seems to make a steric clash with residues on the GSK3β peptide substrate (*Figure 6—figure supplement 2*), which can be an explanation for how Tyr18 contributes to the PH domain-mediated autoinhibition. Thus, we hypothesized that this putative Tyr18-Phe309 interdomain interaction could prevent an Akt substrate from having a favorable orientation for the phosphoryl transfer reaction, thus autoinhibiting the enzymatic activity of Akt.

To test the potential contribution of Phe309 in kinase activity regulation, we proceeded with site-directed mutagenesis. Our initial attempt at generating F309A Akt was hampered by the fact that Thr308 phosphorylation by PDK1 proved difficult with this mutant. We then tried F309L, as the AGC

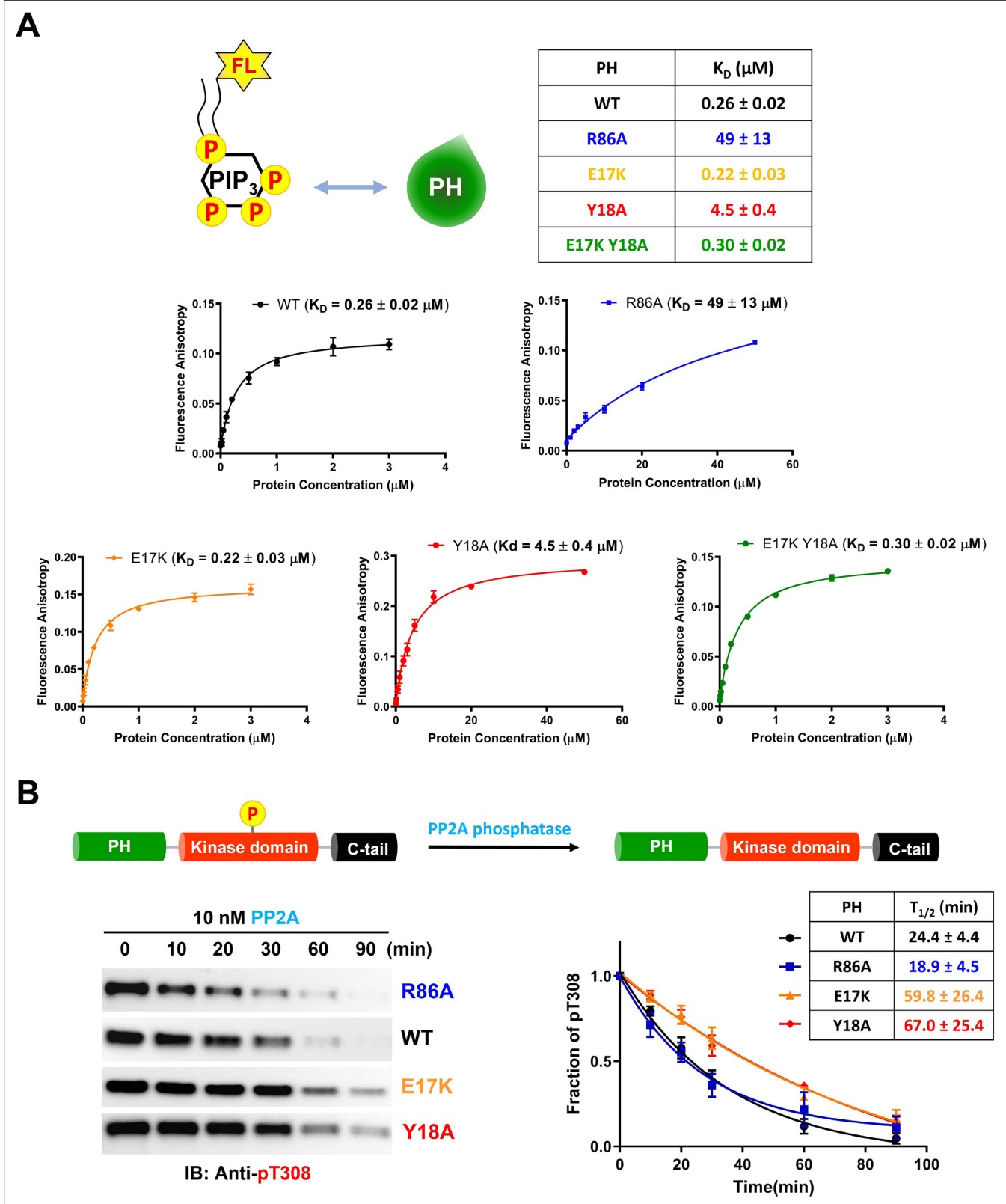

**Figure 5.** Phospholipid, phosphatidylinositol 3,4,5-triphosphate (PIP3) binding affinities and PP2A-mediated phosphatase removal of pT308 among several full-length Akt mutant forms. (**A**) PIP3 binding assays with several isolated Akt PH domain forms. WT: black, R86A: blue, E17K: yellow, Y18A: red, E17K Y18A: green. Fluorescent anisotropy spectra were obtained from the mixture of 50 nM fluorescein-labeled soluble (C8) PIP3 and varying amount of Akt PH domain (0–50 μM). The $K_D$ values were determined from fluorescence anisotropy versus PH domain protein concentration plots (n=3, SD shown).

*Figure 5 continued on next page*

*Figure 5 continued*

(**B**) Full-length Akt mutants have different sensitivities toward the dephosphorylation of pT308 by PP2A phosphatase. Dephosphorylation assays were performed with Akt mutants containing pT308 and lacking C-tail phosphorylation (A6–A9). The dephosphorylation rates were monitored by western blots with anti-pT308 antibody and $T_{1/2}$ values shown were determined from plots for fraction of pT308 versus time (n≥3, SEM shown).

The online version of this article includes the following source data and figure supplement(s) for figure 5:

**Source data 1.** Fluorescence anisotropy binding experiments to measure the phospholipid, phosphatidylinositol 3,4,5-triphosphate (PIP3) binding affinities of the PH domain mutants.

**Source data 2.** Dephosphorylation assays using PP2A phosphatase on full-length Akt mutants containing pT308 and a non-phosphorylated C-tail with raw western blot images.

**Figure supplement 1.** E17K mutation on the PH domain enhances its phosphatidylinositol 4,5-bisphosphate (PIP2) binding affinity while not changing sensitivity toward dephosphorylation of pS473.

**Figure supplement 1—source data 1.** Fluorescence anisotropy binding experiments to measure the phosphatidylinositol 4,5-bisphosphate (PIP2) binding affinities of the PH domain mutants.

**Figure supplement 1—source data 2.** Dephosphorylation assays using alkaline phosphatase on full-length Akt mutants having pT308 and pS473 with raw western blot images.

kinase PKA that is also phosphorylated by PDK1 carries a Leu at the corresponding position in its activation loop (***Chan et al., 1999***). We demonstrated that F309L Akt could be phosphorylated efficiently on Thr308 by PDK1 using western blot analysis (***Figure 6—figure supplement 3***). We prepared both full-length F309L Akt as well as PH-deleted F309L Akt lacking C-tail phosphorylation (***Figure 6—figure supplement 4***) and measured their kinase activities. The catalytic activity of the full-length F309L Akt (A14, V/E 2.33 min$^{-1}$) was found to be approximately twofold greater than WT Akt (A6, V/E 1.26 min$^{-1}$; ***Figure 6B***). In contrast, PH-deleted F309L Akt (A15, V/E 12 min$^{-1}$) was demonstrated to be 40% lower compared with WT PH-free Akt (A16, V/E 19 min$^{-1}$; ***Figure 6C***), resulting in 15-fold and 5-fold reduction of the kinase activity by the presence of the PH domain in WT and F309L Akt, respectively (***Figure 6D***). These results suggest that the F309L replacement modestly reduces the catalytic activity of the free kinase domain but confers activation in the full-length, presumably because of partial relief of autoinhibition.

To examine the potential for a direct interdomain interaction between Phe309 on the kinase domain and Tyr18 on the PH domain of Akt, we pursued an $^{19}$F NMR-based approach. Application of a recently developed method to efficiently incorporate fluoro-Tyr at the aa18 position of the Akt PH domain having R86A and Phe mutations at two of its three natural Tyr, along with the three-piece EPL strategy, allowed us to prepare full-length Akt forms having single fluoro-Tyr at the desired location (***Boeszoermenyi et al., 2019***; ***Chu et al., 2020***). Using this approach, we prepared F309L and Phe309 Akt containing fluoro-Tyr18 in the presence or absence of pT308, as well as the isolated R86A PH domain with fluoro-Tyr18 as a control to determine the NMR spectrum of fluoro-Tyr18 in the absence of the kinase domain of Akt. The fluorine is ortho to the hydroxyl on the phenol side chain of Tyr.

Interestingly, the 1D $^{19}$F NMR spectrum on the isolated R86A PH domain showed two prominent peaks for a single fluoro-Tyr18 and the relative peak intensities changed at different temperatures, implying that the side chain of Tyr18 in the isolated PH domain adopts multiple exchanging conformations (***Figure 7—figure supplement 1***). As the $^{19}$F resonance at −137 ppm was the largest, we assigned this signal to the primary conformation of Tyr18 in the isolated R86A PH domain and the second largest $^{19}$F peak around −136.2 ppm to the principal secondary conformation in this context. The second conformation could also be due to a ring flip since the presence of $^{19}$F makes the ring flip asymmetric.

The $^{19}$F NMR spectrum of Phe309 full-length Akt with pT308 showed a single large and broad peak around −136 ppm (***Figure 7A***), suggesting that fluoro-Tyr18 on the PH domain interacts with the kinase domain of Akt and adopts one major conformation that could be similar to the secondary conformation/ring flip discussed above for the isolated PH domain. This could reflect conformational selection. It is plausible that this fluoro-Tyr −136 ppm chemical shift similarity between the secondary peak of the isolated PH domain and the corresponding signal in the full-length Akt protein is coincidental, arising from a composite of environmental effects. Regardless, the fluoro-Tyr18 peak in the full-length Akt is influenced by the intramolecular interaction of the PH domain with the kinase domain. NMR analysis of fluoro-Tyr containing Phe309 Akt lacking Thr308 phosphorylation showed an upfield

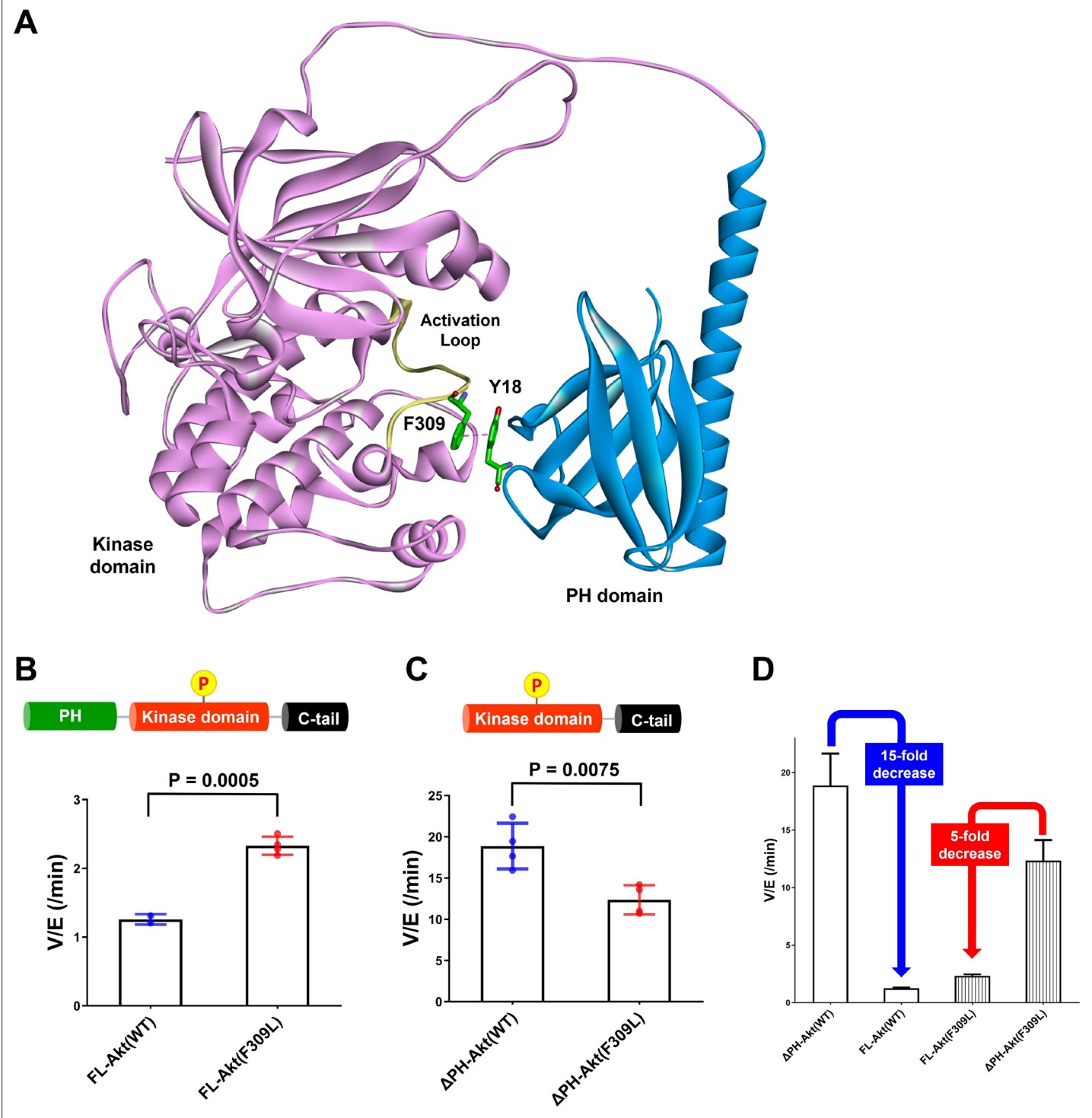

**Figure 6.** Investigating the role of Phe309 in PH domain-mediated Akt autoinhibition. (**A**) Proposed interaction between Tyr18 of the PH domain and Phe309 on the activation loop of the Akt kinase domain in the AlphaFold-predicted full-length Akt structure. (**B**) Enzymatic activities of full-length WT (A6) or F309L (A14) Akt mutant forms containing pT308 and non-phosphorylated C-tails. (**C**) Enzymatic activities of PH domain-deleted WT or F309L Akt mutant forms containing pT308 and non-phosphorylated C-tails. (**D**) The enzymatic activities of WT (A16) or F309L (A15) Akt mutant forms conferred by the PH domain. Kinase assays were performed with each Akt construct in a buffer containing 250 µM ATP and 20 µM GSK3 peptide as substrates (n = 4, SD shown).

The online version of this article includes the following source data and figure supplement(s) for figure 6:

**Source data 1.** Kinase activity assays with full-length or PH domain-deleted Akt mutant forms (WT, F309L) having pT308 and non-phosphorylated C-tail.

*Figure 6 continued on next page*

*Figure 6 continued*

**Figure supplement 1.** The AlphaFold-predicted full-length Akt structure with its confidence score (pLDDT).

**Figure supplement 2.** The proposed F309-Y18 interaction would sterically clash with residues on an Akt peptide substrate.

**Figure supplement 3.** The F309L Akt mutant is efficiently phosphorylated at Thr308 via in vitro by PDK-1.

**Figure supplement 3—source data 1.** Raw western blot images showing the efficient phosphorylation of the F309L mutant at Thr308 by PDK-1.

**Figure supplement 4.** Validation of modified Akt proteins.

shift of the fluoro-Tyr peak relative to pT308 Akt form (**Figure 7B**). This more upfield chemical shift in the direction of the fluoro-Tyr peak of the isolated PH domain suggests that the fluoro-Tyr18 is less effectively locked in the restricted conformation in the absence of pT308. More broadly, these findings are consistent with the idea that the activation loop may directly engage with Tyr18.

As shown in **Figure 7C**, the replacement of Phe309 with Leu also induces upfield shifting of the fluoro-Tyr NMR peak in the setting of pT308. Moreover, in the absence of phosphorylation at Thr308, the F309L drives the $^{19}$F peak even more upfield, approaching that of the $^{19}$F signal of the isolated PH domain (**Figure 7D**). Taken together, the $^{19}$F NMR studies on the different Akt forms are consistent with the notion that Tyr18 and Phe309 are making contact in the autoinhibited state.

### Cellular analysis of Y18A Akt

To test Tyr18-mediated autoinhibition of Akt in a cellular context, we conducted transfection experiments in HCT116 Akt1/2 KO cells. Since the Y18A mutant displays reduced PIP3 binding affinity, we employed N-terminally myristoylated Akt (Myr-Akt) mutants that constitutively localize to the plasma membrane regardless of PH-mediated PIP3 binding status (**Kohn et al., 1996**). This modification was achieved by incorporating a myristoylation signal sequence at the N-terminus of each Akt mutant that can be recognized and myristoylated by an N-myristoyltransferase in a cellular context (**Kohn et al., 1996**). Along with the incorporation of myristoylation for equal activation at the plasma membrane, we also introduced Ala mutation at Ser473 (S473A) for each Akt mutant to determine how the mutation on the PH domain affects the Akt activity by altering the inhibitory PH-kinase interdomain interaction while excluding the effect of pS473.

The cellular activity of Akt was determined by the phosphorylation level of Akt substrate Foxo1/3a (p-Foxo1/3a) using western blot analysis. As expected, non-Myr WT Akt showed just comparable p-Foxo1/3a level with that of non-transfected control, and the incorporation of myristoylation signal sequence on WT Akt was effective in increasing its phosphorylation level at pT308 (**Figure 7E**). Despite this elevated pT308, Myr-Akt WT showed only a minor increase in p-Foxo1/3a relative to non-Myr Akt, suggesting that in the absence of pS473, Akt with pT308 is still effectively autoinhibited by the PH domain. On the other hand, we observed a sharp increase in p-Foxo1/3a induced by Myr-Akt Y18A mutant compared to that of Myr-Akt WT (**Figure 7F**). These results indicate that mutation of Tyr18 to Ala relieves the PH domain-mediated autoinhibition and renders Myr-Akt hyperactive in a cellular context, even without C-tail phosphorylation at Ser473.

### Discussion

Capturing a detailed picture of the autoinhibited state of Akt has remained a challenge for over two decades (**Chu et al., 2020**). As discussed, the existing X-ray structures of near full-length Akt depict inhibited conformations but their physiological relevance is uncertain based on their inclusion of small molecules, nanobodies, or PH-kinase domain linker alterations (**Ashwell et al., 2012**; **Quambusch et al., 2019**; **Truebestein et al., 2021**; **Wu et al., 2010**). These structures appear to show occlusion of the PIP3 binding pocket of the PH domain, although autoinhibited Akt shows similar affinities for soluble PIP3 compared with activated Akt (**Chu et al., 2018**; **Cole et al., 2019**). Moreover, as even the pT308 form of Akt lacking C-terminal phosphorylation is autoinhibited by PH-kinase interactions, and the prior crystal structures lack activation loop phosphorylation, crystallography has been limited as a tool to understand these conformational states (**Chu et al., 2020**; **Cole et al., 2019**).

Intending to characterize how the non-canonical pS477/pT479 activates Akt, we stumbled on a critical hydrogen bond within the PH domain that govern its stability and affinity for the kinase domain. Paradoxically, breaking the Arg86-Glu17 hydrogen bond by R86A mutation sharply increases

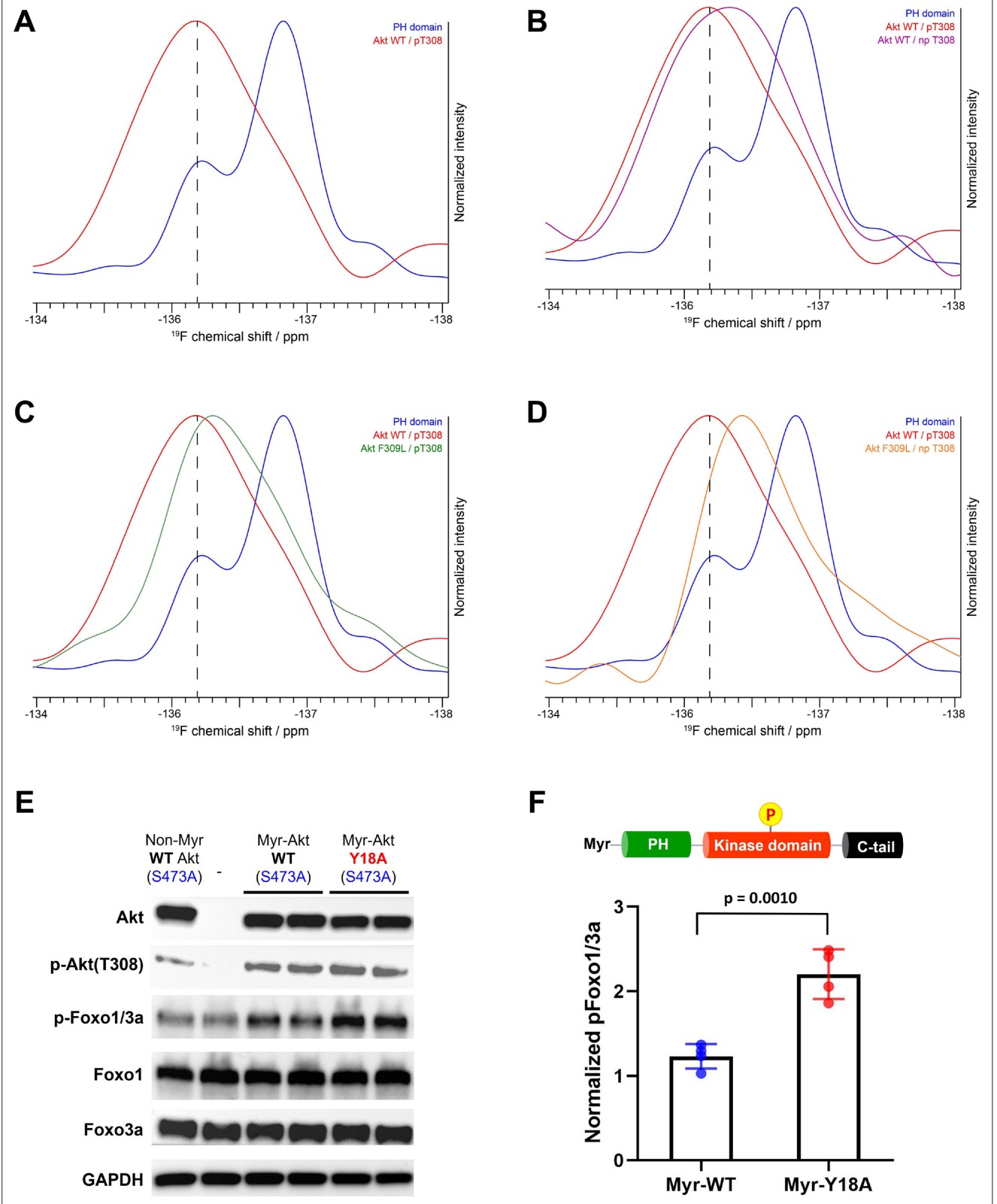

**Figure 7.** NMR analysis of fluoro-Tyr18-containing Akt forms and cellular assays on myristoylated Y18A Akt. (**A–D**) $^{19}$F NMR spectra acquired on the isolated PH domain (blue), and full-length Akt WT (red) and F309L with pT308 (green), as well as full-length Akt WT (purple) and F309L with no pT308 (orange). The PH domain is specifically labeled with $^{19}$F at position 3 of Tyr18 aromatic ring and includes the R86A mutation. All spectra are normalized in intensity to the highest signal (major conformation in isolated PH domain). (**E**) Cellular analysis of the effects of Y18A mutation on Akt activity.

*Figure 7 continued on next page*

*Figure 7 continued*

Akt1/2 knockout HCT116 cells were transfected to express non-Myr WT Akt, myristoylated Akt (Myr-Akt) WT, or Myr-Akt Y18A mutant carrying an S473A mutation and then western blot analysis was done with Akt, Foxo1, and Foxo3a primary antibodies. (**F**) Quantified p-Foxo1/3a levels from cells expressing Myr-WT or Y18A Akt. The p-Foxo1/3a levels are normalized based on that of non-Myr WT Akt (n=4, SD shown).

The online version of this article includes the following source data and figure supplement(s) for figure 7:

**Source data 1.** Quantification of cell-based assays with raw western blot images.

**Figure supplement 1.** [19]F NMR spectra acquired on the isolated PH domain incorporating fluoro-Tyr at Y18 position at different temperatures.

PH domain stability and promotes a Tyr18 rotamer which enhances the binding of the PH domain to the kinase domain. It is the phenyl ring rather than the hydroxyl that mediates its interactions with the kinase domain. Although our studies do not provide definitive information on how pS477/pT479 activates Akt, it is still possible that these phosphorylations may do so by directly engaging Arg86.

Our investigation of the role of Arg86 has also led to a deeper understanding of the molecular mechanisms associated with the well-known Akt E17K oncogenic mutant. At first glance, it appears paradoxical that replacing Glu17 with Lys would be activating, since, like R86A, this mutation disrupts the Glu17-Arg86 salt bridge, and as discussed R86A promotes autoinhibition. We believe the key difference relates to the structural properties of Lys17. In a previously determined X-ray crystal structure of the E17K Akt PH domain, the Lys17 side chain makes a hydrogen bond with the Tyr18 phenol and restrains the Tyr side chain from engaging the Akt kinase domain (*Carpten et al., 2007*). Thus, in our model, the E17K mutation exerts its functional impact indirectly through its effects on the lead player Tyr18. Moreover, we now identify three reasons why E17K Akt is activated in cell signaling. First, we confirm the enhanced affinity of the E17K PH domain for PIP2 which could encourage its recruitment to membranes, as posited previously (*Landgraf et al., 2008*). Second, there is an increase in E17K Akt catalytic activity when the C-tail is non-phosphorylated which would allow for Akt-mediated protein substrate phosphorylation in the absence of mTORC2 action. It should be noted that prior studies that concluded that E17K provided no direct catalytic advantage appeared to measure E17K Akt catalytic activity with C-terminally phosphorylated status (*Carpten et al., 2007*). Such C-terminal phosphorylation would mitigate the stimulatory catalytic effects of E17K, as seen in our own kinetic analysis. Third, pT308 in E17K Akt is resistant to PP2A-mediated dephosphorylation which could prolong Akt activation. We propose it is the combination of each of these three factors which likely renders E17K Akt a potent oncogene.

A combination of enzymology, mutagenesis, AlphaFold, and fluorine NMR experiments point to an unexpected model in which Tyr18 participates in a stabilizing interaction with the side chain of Phe309. The AlphaFold-predicted structure appeared unlikely initially since the activation loop containing Phe309 is commonly considered quite flexible and the prediction for the PH-kinase interdomain interface is limited (*Figure 6—figure supplement 1*). However, our experimental efforts to test hypothesis developed based on the AlphaFold-predicted Akt structure suggested the possibility that Phe309 on the activation loop engages in an autoinhibitory interdomain interaction between Tyr18 and the kinase domain. It is also plausible that the Tyr18-Phe309 interaction allows nearby hydrophobic residues, such as Leu321 to engage in the hydrophobic interaction between the PH and kinase domains, which would be consistent with the AlphaFold prediction and nanobody/Akt crystal structure (*Truebestein et al., 2021*). Given the limitations of the approaches described here, future additional work will ultimately be needed to confirm the detailed PH-kinase domain interface. Furthermore, it will be also interesting in the future to clearly elucidate how differential C-tail modification status affects the key interactions at the PH-kinase domain interface governing the PH domain-mediated autoinhibition of Akt.

Our approach to generating specific, chemically modified Akt forms has involved EPL to generate site-specific C-tail modified forms of Akt. Although it has been suggested by Truebestein et al. that EPL may result in unstable forms of Akt due to a lack of Thr450 phosphorylation (*Truebestein et al., 2021*), our recent studies have not substantiated this concern and confirmed that our semisynthetic Akt constructs contain comparable levels of T450 phosphorylation to commercial fully recombinant full-length Akt (*Salguero et al., 2022*). Indeed, one of our semisynthetic phospho-Akts produced using EPL shows a nearly identical X-ray structure to the fully recombinant S473D Akt (*Chu et al.,*

*2018*). We hypothesize that Truebestein et al.'s contradictory findings may have emanated from the use of a shortened and unnatural PH-kinase linker in their Akt construct.

In summary, through a combination of structural and biochemical approaches, we have developed a new model for how Akt is autoinhibited by its PH domain and how E17K relieves such autoinhibition. It is hoped that the mechanistic insights here may pave the way toward novel targeted inhibitors that can exploit the distinct features of E17K Akt that can be used for the treatment of cancer.

# Materials and methods

## Key resources table

| Reagent type (species) or resource | Designation | Source or reference | Identifiers | Additional information |
|---|---|---|---|---|
| Gene (human) | Akt1 | Addgene https://doi.org/10.1093/nar/gkh238 | https://identifiers.org/RRID/RRID:addgene_9021 | |
| Strain, strain background (*Escherichia coli*) | Rosetta 2 (DE3) pLysS | Novagen | Cat. No.: 71400 | Competent cells |
| Strain, strain background (*Escherichia coli*) | DH10Bac | Invitrogen | Cat. No.: 18297010 | Competent cells |
| Cell line (insect cell) | Sf21 | Invitrogen | 11497-013 | |
| Cell line (insect cell) | Sf9 | Invitrogen | 11496-015 | |
| Cell line (human) | Akt1/2 KO HCT116 | https://doi.org/10.1073/pnas.0914018107 | | The Akt1$^{-/-}$and Akt2$^{-/-}$ HCT116 colon cancer cell line was given as a gift from Dr Bert Vogelstein (Johns Hopkins University) (*Ericson et al., 2010*). These cells were validated by western blot showing the absence of Akt and by the lack of signaling response to growth factors. They were also shown to be mycoplasma-free by PCR testing. |
| Antibody | pan-Akt (11E7) (Rabbit monoclonal) | Cell Signaling Technology | Cat. No.: 4685S, RRID:AB_10698888 | WB (1:1000) |
| Antibody | Akt phospho-Thr308 (Rabbit monoclonal) | Cell Signaling Technology | Cat. No.: 9275S, RRID:AB_329828 | WB (1:1000) |
| Antibody | Akt phospho-Ser473 [EP2109Y] (Rabbit monoclonal) | Abcam | Cat. No.: ab81283, RRID:AB_2224551 | WB (1:1000) |
| Antibody | Foxo1 (C29H4) (Rabbit monoclonal) | Cell Signaling Technology | Cat. No.: 2880S, RRID:AB_2106495 | WB (1:1000) |
| Antibody | Foxo3a (75D8) (Rabbit monoclonal) | Cell Signaling Technology | Cat. No.: 2880S, RRID:AB_836876 | WB (1:1000) |
| Antibody | Phospho-Foxo1(Thr24)/Foxo3a(Thr32) (Rabbit monoclonal) | Cell Signaling Technology | Cat. No.: 2880S, RRID:AB_329842 | WB (1:1000) |
| Antibody | GAPDH (14C10) (Rabbit monoclonal) | Cell Signaling Technology | Cat. No.: 2118S, RRID:AB_561053 | WB (1:5000) |
| Antibody | HRP conjugated, anti-Rabbit IgG (Goat monoclonal) | Cell Signaling Technology | Cat. No.: 7074S, RRID:AB_2099233 | WB (1:5000) |
| Chemical compound, drug | Ammonium chloride (15N, 99%) | Cambridge Isotope Laboratories | Cat. No.: NLM-467-1 | |
| Chemical compound, drug | Glyphosate | Sigma-Aldrich | Cat. No.: 337757 | |
| Chemical compound, drug | L-Phenylalanine | Sigma-Aldrich | Cat. No.: P5482 | |

*Continued on next page*

*Continued*

| Reagent type (species) or resource | Designation | Source or reference | Identifiers | Additional information |
|---|---|---|---|---|
| Chemical compound, drug | L-Tryptophan | Sigma-Aldrich | Cat. No.: T0254 | |
| Chemical compound, drug | 3-Fluoro-L-Tyrosine | AmBeed | Cat. No.: A374537 | |
| Chemical compound, drug | Sulfo-Cy5-NHS ester | Lumiprobe | Cat. No.: 43320 | |
| Chemical compound, drug | 32P-ATP | Perkin Elmer | Cat. No.: NEG002Z2-50UC | |
| Chemical compound, drug | PIP3-Fluorescein (ammonium salt) | Cayman Chemical | Cat. No.: 10010383 | |
| Chemical compound, drug | PIP2-Fluorescein (ammonium salt) | Cayman Chemical | Cat. No.: 10010388 | |
| Chemical compound, drug | Pierce Avidin | Thermo Fisher Scientific | Cat. No.: 21128 | |
| Chemical compound, drug | PreScission protease | Cytiva | Cat. No.: 27-0843-01 | |
| Chemical compound, drug | Alkaline phosphatase (Calf Intestinal) | NEB | Cat. No.: M0290S | |
| Chemical compound, drug | PP2A catalytic subunit (human recombinant, L309 deletion) | Cayman Chemical | Cat. No.: 10011237 | |
| Software, algorithm | Coot | https://doi.org/10.1107/s0907444910007493 | | |
| Software, algorithm | ImageQuant TL version 7.0 | GE Healthcare | | |
| Software, algorithm | GraphPad Prism version 9.1.2 | GraphPad | | |
| Software, algorithm | NmrPipe | https://doi.org/10.1007/BF00197809 | | |
| Software, algorithm | CCPNmr Analysis version 2.4 | https://doi.org/10.1002/prot.20449 | | |
| Software, algorithm | Topspin version 3.6 | Bruker | | |

## Semisynthesis of Akt

The baculovirus-insect cell expression system was employed to express Akt-*Mxe*intein-CBD (CBD: chitin binding domain) constructs (Akt1 aa 1–459 and 122–459) according to the previously reported procedures (*Chu et al., 2018*). To produce full-length Akt containing mutation on its PH domain and the C-tail site-specific phosphorylations at either Ser473, Ser477/Thr479, or no phosphorylation on these residues, a three-piece EPL strategy was employed as previously described (*Chu et al., 2020*). Briefly, after resuspending the *Escherichia coli* cells expressing N-terminally His-tagged PH domain-*Mxe*Intein-CBD in lysis buffer (50 mM HEPES pH 7.5, 150 mM NaCl, 1 mM EDTA, 10% glycerol, 0.1% Triton X-100, 1 mM PMSF, one protease inhibitor tablet [Thermo Fisher Scientific]), the cells were lysed by french press and the resulting lysate was cleared by centrifugation at 15,000× *g* for 40 min at 4°C. The insect cells expressing Akt (S122C, aa 122–459)-*Mxe*Intein-CBD were resuspended in lysis buffer and lysed in a 40 mL Dounce homogenizer on ice, and the resulting lysate was cleared by centrifugation at 15,000× *g* for 40 min at 4°C. Next, both N-Tags-TEV-S122C-Akt kinase domain (aa 122–459)-*Mxe*Intein-CBD (N-tags: N-terminal Flag-HA-6xHis) and N-terminally His-tagged Akt PH domain (aa 1–121)-*Mxe*Intein-CBD proteins were purified by chitin bead-mediated affinity chromatography from the cell lysates. The protein C-terminal thioester forms of both the Akt PH and kinase

domains were eluted via intein cleavage using MESNA (sodium mercaptoethylsulfonate) according to established protocols (*Chu et al., 2018*).

The resulting N-Tags-TEV-S122C-Akt kinase domain thioester was phosphorylated at Thr308 via in vitro phosphorylation reaction by recombinant GST-PDK1 and then ligated with the synthetic N-Cys containing C-terminal Akt peptides (aa 460–480) having variable phosphorylation in the first ligation buffer (50 mM HEPES, pH 7.5, 150 mM NaCl, 1 mM TCEP, 100 mM MESNA, 10 mM EDTA, 10% glycerol, 1 mM PMSF) for 5 hr at room temperature and then maintained overnight at 4°C. The preparation of the recombinant GST-PDK1 and the synthetic C-terminal Akt peptides were described in our previous work (*Chu et al., 2018*; *Chu et al., 2020*). The ligation product N-Tags-TEV-Akt (S122C, aa 122–480) was purified by size exclusion chromatography (SEC) on a Superdex75 10/300 GL column (GE Healthcare) equilibrated with the second ligation buffer (50 mM HEPES, pH 7.5, 150 mM NaCl, 100 mM MESNA, 1 mM TCEP, 1 mM PMSF). The pure fractions (>90%) were combined, concentrated, and mixed with highly concentrated Akt PH domain thioester (5 mg/mL). After that, 15 µg of TEV protease was added to cleave the N-terminal tags and expose the first cysteine residue to initialize the second EPL reaction at 4°C for 72 hr. The ligation yields were determined by SDS-PAGE followed by Coomassie staining and were typically greater than 90%. The remaining unligated kinase domain was removed using the His-tag purification method as previously described (*Spriestersbach et al., 2015*). After cleaving the N-terminal His-tag by adding 15 µL of PreScission Protease (Cytiva) while performing overnight dialysis at 4°C against Akt storage buffer (50 mM HEPES, pH 7.5, 150 mM NaCl, 1 mM EDTA, 2 mM beta-mercaptoethanol, 10% glycerol), the desired full-length Akt (aa 1–480) was separated from the excess of Akt PH domain thioester by using SEC purification on a Superdex200 10/300 GL column (GE Healthcare) equilibrated with Akt storage buffer, and then concentrated, flashfrozen, and stored at –80°C. Previously described two-piece EPL strategy with the intein-mediated Akt (aa 1–459)-thioester fragment and the synthetic N-Cys containing C-terminal Akt peptides (aa 460–480) was also employed to make Akt mutant proteins having differentially phosphorylated C-tails (*Chu et al., 2018*).

## Semisynthesis of labeled Akt for NMR studies

To obtain the $^{15}$N-labeled Akt PH domain, the Akt (aa 1–121)-*Mxe*Intein-CBD was expressed in *E. coli* Rosetta (DE3)/pLysS (Novagen) following the established protocol (*Coote et al., 2018*; *Gronenborn et al., 1991*). Briefly, the *E. coli* cells were grown in 1 L of M9 minimal medium (6 g/L $Na_2HPO_4$, 3 g/L $KH_2PO_4$, 0.5 g/L NaCl, 0.25 g/L $MgSO_4$, 11 mg/L $CaCl_2$, 5 g/L C-glucose, 1 g/L $^{15}NH_4Cl$ (Cambridge Isotopes), 100 mg/L ampicillin and 20 mg/L chloramphenicol) in $H_2O$, and was further supplemented with trace elements (50 mg/L EDTA, 8 mg/L $FeCl_3$, 0.1 mg/L $CuCl_2$, 0.1 mg/L $CoCl_2$, 0.1 mg/L $H_3BO_3$, and 0.02 mg/L $MnCl_2$) and the vitamins biotin (0.5 mg/L) and thiamin (0.5 mg/L) in shaker flasks at 37°C until $OD_{600}$=0.5. After that, IPTG was added to make a final concentration of 0.5 mM to induce the expression and the culture was further incubated for 18 hr at 16°C. The cells were pelleted and stored in –80°C freezer for the next steps. To incorporate fluoro-Tyr into the Akt PH domain, the Akt (aa 1–121, Y26F, Y38F, R86A)-*Mxe*Intein-CBD was expressed in *E. coli* Rosetta (DE3)/pLysS (Novagen) as previously described (*Boeszoermenyi et al., 2019*). Briefly, the *E. coli* cells were grown in 1 L of M9 minimal medium (6 g/L $Na_2HPO_4$, 3 g/L $KH_2PO_4$, 0.5 g/L NaCl, 0.25 g/L $MgSO_4$, 11 mg/L $CaCl_2$, 5 g/L C-glucose, 1 g/L $NH_4Cl$, 100 mg/L ampicillin, and 20 mg/L chloramphenicol) in $H_2O$, and was further supplemented with trace elements (50 mg/L EDTA, 8 mg/L $FeCl_3$, 0.1 mg/L $CuCl_2$, 0.1 mg/L $CoCl_2$, 0.1 mg/L $H_3BO_3$, and 0.02 mg/L $MnCl_2$) and the vitamins biotin (0.5 mg/L) and thiamin (0.5 mg/L) in shaker flasks at 37°C until $OD_{600}$=0.5. After that, 1 g of glyphosate, as well as 50 mg of phenylalanine, tryptophan, and 3-fluoro-tyrosine were added. When $OD_{600}$ = 0.7, 1 mL of 0.5 M IPTG was added to induce the expression, and the culture was further incubated for 18 hr at 16°C. The cells were pelleted and stored in –80°C freezer for the next steps. The three-piece EPL strategy described above was employed to produce full-length Akt containing labeled PH domain ($^{15}$N or fluoro-Tyr18) for NMR studies.

## NMR experiments

If not stated otherwise, NMR experiments were conducted on a Bruker spectrometer operating at 750 MHz, equipped with a triple-channel $^1$H, $^{13}$C, $^{15}$N TCI cryoprobe, and z-shielded gradients. All data were processed using Topspin (Bruker) or NmrPipe (*Delaglio et al., 1995*) and analyzed with CCPNmr

(*Vranken et al., 2005*). For NMR resonance assignments, samples of approximately 600 µM isotopically labeled R86A Akt PH domain (aa 1–121) in 25 mM HEPES pH 6.5, 500 mM NaCl, 1 mM TCEP, and 5% v/v $^2H_2O$ were measured at 13°C. Due to the high ionic strength, rectangular-shaped sample tubes were used and effectively reduced the $^1H$ 90° pulse length and increased sensitivity significantly (*de Swiet, 2005*). Eighty-one percent of non-proline backbone resonances were assigned, significantly more than for the WT counterpart due to the apparent higher stability and rigidity of the R86A PH domain (*Chu et al., 2020*). Chemical shift assignments were deposited in the BMRB under accession code 51419. HSQC spectra of 100–120 µM segmentally isotopically labeled full-length R86A Akt were measured in 25 mM HEPES pH 6.5, 500 mM NaCl, 1 mM TCEP, and 5% v/v $^2H_2O$ using TROSY-HSQC. Recycle delays of 1 s were used and approximately 768 transients (and 256 increments) were accumulated, leading to measurement times of 54 hr. Chemical shifts were calibrated using internal water calibration in NmrPipe. Combined CSPs and associated standard deviations were calculated according to the protocol from *Schumann et al., 2007*.

Nuclear spin relaxation ($R_1$, $R_2$, $Eta_{xy}$) were measured on a Bruker 800 MHz using standard pulse sequences and recycle delays of 5 s. $R_2$ relaxation employed Carr-Purcell-Meiboom-Gill (CPMG) pulse trains with varied delays and corresponding temperature compensation blocks. All experiments were acquired using single scan interleaving. $R_2$ relaxation delays were set to 10, 30, 50, 70, 90, 130, 150, 170 ms (50 ms was repeated twice). $R_1$ relaxation delays were set to 0.01, 0.05, 0.1, 0.2, 0.3, 0.5, 0.8, 1, 1.2, 1.5, 1.8 s (0.5 s was repeated twice). Cross-correlated (DD/CSA) relaxation rates were measured at a relaxation delay of 50 ms. For steady-state hetNOE, $^1H$ saturation was achieved with a train of 180° pulses at 12 kHz power for a delay of 6 s (>4·$T_1^{max}$), recycle delays of 10 s were employed. CPMG relaxation dispersion experiments were conducted with a $T_2$ relaxation delay of 40 ms. CPMG frequencies were modulated in the following order (in Hz): 0, 1000, 50, 850, 100, 650, 500, 250, 400, 100, 50, 850, 400, 1000. Dispersion curves were analyzed with the program Relax.

$^{19}F$ experiments were measured on a Bruker spectrometer operating at 600 MHz, equipped with a QCIF triple-channel $^1H$, $^{19}F$, $^{13}C$, $^{15}N$ TCI cryoprobe, and z-shielded gradients. Samples of approximately 25–50 µM R86A Akt PH domain modified with 3-$^{19}F$-Tyr18 were measured, either isolated or after EPL with Akt kinase domain. Two other Tyr residues of the PH domain, Tyr26, and Tyr38, were mutated to Phe, hence the $^{19}F$ signal can be assigned to Tyr18. $^{19}F$ 1D spectra were measured with relaxation delays of 1 s, 1024 points in the direct dimension for a spectral width of 200 ppm (4.5 ms acquisition time), and 64k scans for a total experimental time of approximately 10 hr.

## Crystallization of the R86A PH domain followed by data collection and structural determination

The pTXB1 plasmid containing Akt PH domain (aa 1–121, R86A)-*Mxe*Intein-CBD sequence was transformed into *E. coli* Rosetta (DE3)/pLysS (Novagen) to express corresponding protein construct using the established protocol (*Coote et al., 2018*; *Gronenborn et al., 1991*). Briefly, the *E. coli* cells were grown in 1 L of LB medium in a shaker flask at 37°C until OD$_{600}$=0.6, and then IPTG was added to make a final concentration of 0.5 mM to induce the expression and the culture was further incubated at 16°C for 18 hr. After that, cells were pelleted and resuspended in lysis buffer (50 mM HEPES pH 7.5, 150 mM NaCl, 1 mM EDTA, 10% glycerol, 0.1% Triton X-100, 1 mM PMSF, one protease inhibitor tablet [Thermo Fisher Scientific]), and then lysed by french press and the lysate was cleared by centrifugation at 15,000× *g* for 40 min at 4°C. Next, the Akt PH domain (aa 1–121, R86A)-*Mxe*Intein-CBD protein was purified by affinity chromatography from the cell lysate using chitin beads. After loading onto chitin beads, the C-terminal thioester form of Akt R86A PH domain was eluted via intein cleavage using 30 mM DTT as previously described (*Chong et al., 1997*), and the resulting protein was incubated at room temperature for 3 days in hydrolysis buffer (50 mM Tris, pH 8.5, 150 mM NaCl, 1 mM TCEP) to make R86A PH protein have the native carboxylic acid group at its C-terminus.

The R86A PH domain protein was concentrated to 10 mg/mL and crystals were grown using the sitting drop method in a 96-well plate by adding a 1:1 volume of well solution (100 mM HEPES, pH 7.5, 1.28 M sodium citrate, 10 mM praseodymium (III) acetate) to protein and incubated at 20°C for 3 days. Crystals were cryoprotected in the same precipitant buffers where they were grown with 20% (v/v) ethylene glycol added, flash-frozen, and stored in liquid nitrogen until data collection.

Diffraction data of R86A PH domain protein were collected at APS NE-CAT 24E on a DECTRIS Eiger X 16 M detector. The data were indexed, integrated, and scaled using XSCALE. The R86A PH

domain crystals were isomorphous with those of the previously reported WT structure (PDB: 1UNP), which was used as an initial model for refinement (*Milburn et al., 2003*). The model was rebuilt manually and validated using Coot (*Emsley et al., 2010*) and PDB Deposition tools. The final structure of the R86A PH domain was refined at 1.39 Å (*Table 1*, PDB ID: 7MYX). Structure figures were prepared with PyMOL (*Delano, 2002*) and Discovery Studio Visualizer.

## Thermal shift assay/differential scanning fluorimetry

WT and R86A PH domains were expressed and purified as previously described. Reactions were prepared in 384-well plates (Greiner Bio-One) in a total volume of 33 µL. SyproOrange dye dilutions in assay buffer (25 mM HEPES pH = 7.5, 150 mM NaCl, 1 mM TCEP) were first prepared by a NT8 liquid handler (Formulatrix) followed by addition of 10 µM PH domain. Dilution series were ×0.2–30 Sypro-Orange dye (molar stock equivalent of ×5000 DMSO commercial stock, Invitrogen). Sealed plates were heated at 1°C/min from 25°C to 95°C with fluorescence reading (586 nm) every 0.5°C in a ViiA-7 real-time PCR system (Applied Biosystems, Thermo Fisher Scientific). Tm values were determined as the minimum of the first derivative of the recorded fluorescence intensity versus temperature. Fluorescence intensity at 25°C and melting temperatures were found independent from dye concentration.

## Intact mass spectrometry analysis

Akt proteins in Akt storage buffer were diluted to 50 ng/µL in 50 mM ammonium bicarbonate (pH = 8.0) and centrifuged at 15,000× *g* for 5 min. Supernatant of each sample was transferred to a low binding liquid chromatography vial, and 100 ng of protein was injected into a Vanquish Flex LC interfaced to a Q Exactive mass spectrometer (Thermo Fisher Scientific). Proteins were separated, ionized, and detected as previously described (*Salguero et al., 2022*). For each protein, mass scans were averaged across the peak in chromatogram, and the resulting spectra were deconvoluted using UniDec software 3.0 (*Marty et al., 2015*).

## Kinase activity assays

The steady-state kinetic parameters of each Akt protein were determined using $^{32}$P-labeled ATP and N-ε-biotin-lysine GSK3 peptide (RSGRARTSSFAEPGGK) as substrates, as previously described with minor modification (*Qiu et al., 2009*). The concentration of the biotinylated GSK3 peptide substrate was fixed at 20 µM (above the peptide substrate $K_m$ values) in all assays. The kinase assay buffer is composed of 50 mM HEPES, pH 7.5, 10 mM $MgCl_2$, 1 mM EGTA, 2 mM DTT, 1 mM sodium orthovanadate, 0.5 mg/mL BSA, approximately 0.5 µCi γ-$^{32}$P-ATP, and varying amounts of ATP (0–2.5 mM). The kinase reactions were performed at 30°C for 10 min and quenched by the addition of 100 mM EDTA, and then 100 µg of Avidin (Pierce) was added to each sample and incubated for 20 min at room temperature. Each sample was transferred to a centrifugal 10 kDa MWCO filtration unit (Nanosep 10 K, PALL) and washed five times with 120 µL of washing buffer (0.5 M sodium phosphate, 0.5 M NaCl, pH 8.5). Each filtration unit containing the biotinylated GSK3 peptide substrate complexed with Avidin was placed in 10 mL scintillation fluid and radioactivity came from the $^{32}$P-labeled phosphate group on the peptide substrate was measured by Beckman liquid scintillation counter (Beckman LS6500) to determine what percentage of the peptide substrate in each sample was phosphorylated in 10 min. Kinetic parameters of each Akt mutant were determined from V/[E] versus [ATP] plots using the Michaelis-Menten equation. As we have noted previously, some of the moderately different kinase activities here versus prior reports may reflect different levels of linker phosphorylations that can vary among insect cell preparations and in vitro versus in cellulo PDK1 phosphorylation used to generate pT308 (*Chu et al., 2020*; *Salguero et al., 2022*). To mitigate the effects of such variability, we compare kinase activities of semisynthetic Akts that are prepared from the same batches of recombinant Akt fragments using identical methods.

## MST binding experiments

For use in MST binding experiments, PH-deleted S122C Akt (aa 122–480, pT308) was N-terminally Cy5 labeled by pretreating Sulfo-Cy5-NHS ester (Lumiprobe) with MESNA to efficiently convert NHS ester into thioester that can react with the N-terminal cysteine selectively, as previously described (*Dempsey et al., 2018*). The binding affinity between the PH-deleted Akt (aa 122–480, pT308) with each Akt PH domain mutant (aa 1–121) in a binding buffer (50 mM HEPES pH 7.5, 150 mM NaCl, 5%

(v/v) glycerol, 0.05% (v/v) Triton X-100, 0.1 mg/mL ovalbumin, 5 mM DTT) was determined by MST binding experiments at 25°C using MONOLITH NT.115 (NanoTemper). Twenty nM of the Cy5-labeled Akt (aa 122–480) was mixed 1:1 with different amounts of each Akt PH domain mutant to prepare samples for the experiments. Each binding assay was repeated three times.

### Fluorescence anisotropy binding experiments

The binding affinity of the Akt PH domain with phospholipid PIP2 or PIP3 was determined using fluorescence anisotropy. Fifty nM fluorescein-labeled soluble (C8) PIP2 or PIP3 (Cayman Chemical) was mixed with varying amounts of Akt PH domain in binding buffer (50 mM HEPES pH 7.5, 2 mM DTT, 0.05 mg/mL ovalbumin) and the resulting mixture was incubated at room temperature for 30 min. Fluorescent anisotropy spectra were recorded by Multi-Mode Microplate Reader (Biotek Instruments) at 23°C with three different replicates. The $K_D$ value for each Akt PH domain mutant was obtained by fitting the data to the quadratic binding equation as described before (*Chu et al., 2018*; *Seamon et al., 2015*; *Weiser et al., 2017*).

### Dephosphorylation assays

The dephosphorylation assays were carried out following the protocol previously described with minor modification (*Bolduc et al., 2013*; *Chan et al., 2011*; *Lin et al., 2012*). Briefly, 500 nM Akt was mixed with either 10 nM PP2A catalytic subunit (309 L deletion, Cayman Chemical) or 20 nM alkaline phosphatase (Calf Intestinal, NEB) in the assay buffer (50 mM HEPES, pH 7.5, 100 mM NaCl, 2 mM DTT, 5 mM MgCl₂) and incubated at 30°C. Ten µL of the reaction was collected at the indicated time points and quenched by the addition of ×2 SDS-loading buffer. The SDS samples were loaded on SDS-PAGE gel for western blot analysis with anti-pT308 (Cell Signaling Technology) or anti-pS473 (Abcam) and pan-Akt (Cell Signaling Technology) primary antibodies.

### Western blots

After transferring proteins from gels to nitrocellulose membranes using an iBlot (Thermo Fisher Scientific) system, 5% (w/v) BSA in TBS-T buffer was used to block the membranes for 30 min incubation at room temperature. After that, the membranes were incubated with primary antibodies (Cell Signaling Technology) at a 1:1000 dilution with 5% (w/v) BSA in TBS-T buffer overnight at 4°C, followed by a 10 min wash three times with TBS-T buffer, and then incubated with secondary HRP-linked secondary antibody (Cell Signaling Technology) for 1 hr at room temperature, followed by 10 min wash three times. Lastly, the membranes were developed with Clarity Western ECL Substrate (Bio-Rad) and imaged by a GeneSys (G:BOX, SynGene) imaging system.

### Mammalian cell signaling assays

The Akt1/2 knockout HCT116 colon cancer cell line was given as a gift from Dr Bert Vogelstein (Johns Hopkins University; *Ericson et al., 2010*). Cells were cultured in McCoy's 5A (Gibco) supplemented with 10% (v/v) FBS (Sigma) and 1% (v/v) penicillin/streptomycin (Gibco) at 37°C and 5% CO₂. When the cell confluence reached around 70% in six-well plates, the medium was changed with McCoy's 5A (Gibco) having 5% (v/v) FBS (Sigma) and 1% (v/v) penicillin/streptomycin (Gibco), and the cells were transfected with 1.5 µg of pcDNA3.1-Flag-HA-Akt plasmid or 1.5 µg (3.0 µg for Y18A mutant) of pcDNA3.1-Myr-HA-Akt plasmid complexed with 3 µL Lipofectamine 3000 (Invitrogen) and 3 µL P3000 reagent (Invitrogen) in Opti-MEM medium (Gibco) at 37°C and 5% CO₂. Seventy-two hours after transfection, the cells were washed with cold PBS and lysed by adding 150 µL RIPA buffer (Cell Signaling Technology) containing ×1 complete protease inhibitor tablet (Thermo Fisher Scientific) and ×1 PhosStop tablet (Roche), and 1 mM PMSF for 10 min at 4°C. Thirty µg of total protein (BCA assay) was loaded on an SDS-PAGE gel. Membrane transfer and western blotting were carried out as described above with 1:1000 dilution for primary antibodies: anti-Akt, anti-pT308 Akt, anti-Foxo1, anti-Foxo3a, and anti-pT24 Foxo1/pT32 Foxo3a, anti-GAPDH (Cell Signaling Technology).

### Acknowledgements

We thank Kwangwoon Lee for helpful advice on structural analysis and we also thank Brad Palanski for helping us in mass spectrometry analysis for our Akt constructs. We acknowledge NCI grant CA74305 (PAC and ALS), and NIH grants R35CA242461 (MJE), R50CA221830 (EP) for generous financial

support. NC was supported by NCI K22 award (K22CA241105). HA is grateful for funding from the Claudia Adams Barr Program for Innovative Cancer Research. Maintenance of the NMR instruments used for this research was supported by NIH grant EB002026 and R01GM136859. HB was supported by the Kwanjeong Educational Foundation pre-doctoral fellowship. This work used NE-CAT beamline (GM124165), an Eiger detector (OD021527) at the APS (DE-AC02-06CH11357).

## Additional information

### Competing interests

Philip A Cole: Senior editor, eLife. The other authors declare that no competing interests exist.

### Funding

| Funder | Grant reference number | Author |
| --- | --- | --- |
| National Cancer Institute | R01CA74305 | Philip A Cole |
| National Cancer Institute | R35CA242461 | Michael J Eck |
| National Cancer Institute | K22CA241105 | Nam Chu |
| Claudia Adams Barr Program | Grant | Haribabu Arthanari |
| National Institutes of Health | EB002026 | Haribabu Arthanari |
| Kwanjeong Educational Foundation | Fellowship | Hwan Bae |
| National Cancer Institute | R50 CA221830 | Eunyoung Park |
| National Institute of General Medical Sciences | R01GM136859 | Haribabu Arthanari |

The funders had no role in study design, data collection and interpretation, or the decision to submit the work for publication.

### Author contributions

Hwan Bae, Conceptualization, Data curation, Formal analysis, Funding acquisition, Validation, Investigation, Methodology, Writing – original draft, Writing – review and editing; Thibault Viennet, Conceptualization, Data curation, Software, Formal analysis, Validation, Investigation, Methodology, Writing – original draft, Writing – review and editing; Eunyoung Park, Conceptualization, Data curation, Software, Formal analysis, Supervision, Funding acquisition, Validation, Investigation, Writing – original draft, Writing – review and editing; Nam Chu, Conceptualization, Data curation, Formal analysis, Supervision, Funding acquisition, Methodology, Writing – review and editing; Antonieta Salguero, Resources, Formal analysis, Funding acquisition, Validation, Investigation, Writing – review and editing; Michael J Eck, Conceptualization, Data curation, Formal analysis, Supervision, Funding acquisition, Investigation, Writing – original draft, Project administration, Writing – review and editing; Haribabu Arthanari, Conceptualization, Data curation, Software, Formal analysis, Supervision, Funding acquisition, Validation, Investigation, Methodology, Writing – original draft, Project administration, Writing – review and editing; Philip A Cole, Conceptualization, Formal analysis, Supervision, Funding acquisition, Methodology, Writing – original draft, Project administration, Writing – review and editing

### Author ORCIDs

Hwan Bae http://orcid.org/0000-0001-5252-252X
Thibault Viennet http://orcid.org/0000-0001-5349-0179
Eunyoung Park http://orcid.org/0000-0001-5618-1267
Michael J Eck http://orcid.org/0000-0003-4247-9403
Haribabu Arthanari http://orcid.org/0000-0002-7281-1289
Philip A Cole http://orcid.org/0000-0001-6873-7824

Decision letter and Author response
Decision letter https://doi.org/10.7554/eLife.80148.sa1
Author response https://doi.org/10.7554/eLife.80148.sa2

## Additional files

### Supplementary files
• MDAR checklist

### Data availability
Diffraction data have been deposited in PDB under the accession code: 7MYX. Source data are uploaded as Zip files on the eLife website.

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
