## [Editor Report]

Bae et al. provide fundamental new insight into molecular features associated with autoinhibited Akt; the Ser/Thr kinase that controls signaling pathways that drive cell survival, proliferation, and cancer. Akt structure/function studies have been ongoing for decades and the data presented in this work convincingly provide support for a new regulatory feature within Akt. As our understanding of Akt regulation is further refined by this and future work in the field, we can anticipate the picture of precisely how this and related kinases are controlled at the molecular level will continue to emerge.

---

## [Decision Letter]

**Decision letter after peer review:**

Thank you for submitting your article "PH Domain-Mediated Autoinhibition and Oncogenic Activation of Akt" for consideration by *eLife*. Your article has been reviewed by 3 peer reviewers, including Amy Andreotti as the Reviewing Editor and Reviewer #1, and the evaluation has been overseen by a Reviewing Editor and Volker Dötsch as the Senior Editor.

Essential revisions:

There are a number of reviewer suggestions to further strengthen this manuscript (see specific reviewer comments below). In particular, two important issues are identified that the reviewers request be addressed in a revised manuscript:

1) Questions are raised about the validity and predictive value of the AlphaFold model. Please see specific reviewer comments below and address them accordingly.

2) Further validation of protein samples is requested. Specifically, MS quantification and gel filtration profiles are requested.

In addition, the framing of this work creates some confusion. Given that this is submitted as a Research Advance, it is framed in the context of different phosphorylations of the C-tail on AKT. Since the results then focus on side chains in the PH domain the framing, as presented, creates some confusion about the goals at the outset and the eventual findings. The reviewers are confident that the authors can reframe the work in a manner that both explains the connection to earlier findings and clearly steers the reader through the present work.

*Reviewer #1 (Recommendations for the authors):*

The following are provided to improve the quality of this manuscript:

1) Figure 3A seems to show multiple peaks for the wt protein possibly indicative of slow exchange (the mutant spectrum appears greatly simplified). The authors do not adequately comment on this or attempt to establish whether the resonances that appear to give rise to multiple peaks are structurally related. Moreover, it is unclear whether the wt sample could be subject to changes in temperature or ZZ-exchange experiments to more carefully analyze the possibility of multiple species in slow exchange. This seems relevant to the idea that the R86A mutation 'stabilizes' the PH domain and in so doing pre-organizes the PH domain for binding to the kinase domain.

2) The notion that the R86A mutation stabilizes the PH domain seems to be based on the longevity of the protein sample. Could a more direct measure of protein stability that does not require the long acquisition times of the NMR relaxation experiments (for example, melting temperature) be used to support this idea?

3) In describing the data for Figure 3—figure supplement 3, the authors state that "we observe similar patterns of CSPs and peak broadening indicative of the interaction of the PH domain with the rest of the protein. Though the comparative CSP patterns of the R86A and WT PH domains when anchored to the rest of Akt look similar, there are subtle differences…". I agree with this conclusion but due to the absence of many peaks in the wt spectrum it is hard to pick out the similarities between the WT and R86A mutant as presented – in fact the histograms in Figure 3—figure supplement 3 look quite different. Perhaps the language used to compare the two datasets can be modified to make the point clearly without creating confusion. Importantly, it seems the authors need to clearly show the differences around residues 67-73.

4) With regard to residues 67-73 it is intriguing that this is where the CSPs seem to differ between wt and R86A. Inspection of the structure shows that R69 in this region is involved in very close interactions with W22 (cation π interaction?). As well, R15 sits on the opposite face of W22 and both R15 and W22 surround the important E17-Y18 segment. Moreover, R69 is surrounded by prolines – could proline isomerization be at play here? Certainly, this is an event that could induce the slow exchange that is suggested in the 1H-15N HSQC spectrum of wt (Figure 3A).

5) In Figure 7A-D. (and Figure 7 supp 1) "PH domain" in blue is the R86A mutant? If so please clarify in the figure and caption – the text indicates the R86A PH domain while the figure is unclear. Assuming Figure 7 is the R86A mutant, was the corresponding data for the wt PH domain also acquired to assess conformational heterogeneity at the Y18 position for wt versus R86A mutant? Such an experiment could help better understand the effects of the R86A mutation on the Y18 sidechain.

*Reviewer #2 (Recommendations for the authors):*

The multi-pronged approach systematically investigates the interaction between PH:Kinase, stability/structure of PH domain, activity of AKT and half-life of activation loop phosphorylation. Methodologically this work is a tour de force with the use of 3-way ligation to generate specific, stoichiometric post-translationally modified variants to study and highlights the limitations of studies using non-physiological variants.

This paper is written for an AKT expert and may not successfully connect with the broader *eLife* audience. The main conclusion of the paper comes quite late and was unanticipated. While such findings are exciting, I found it hard to connect the earlier and later results into a coherent finding.

– This paper is written for an expert and as such may not successfully connect with the broader *eLife* audience. For example, an introduction to the regions of the PH domain responsible for lipid binding and those proposed to bind AKT would help provide context for understanding the early mutagenesis. Perhaps figure 1B could be relabeled to identify these surfaces of the PH domain? Similarly, introducing the models of AKT autoinhibition earlier would help provide a framework for the subsequent experiments.

– The paper is framed as a study in the role of different phosphorylations of the C-tail on AKT but ends with a result that is separate from the tail. At the end of the paper, I'm still pondering how the tail could/would influence this new autoinhibitory interaction. Can the authors clarify how the phosphorylation of the AKT tail influences interactions/activities of variants tested and how those states would influence their model? Relatedly, several times throughout the paper I struggled to determine whether the variant used had the tail present and, if so, its phospho-state. I'm still not sure if the experiments in Figure 2C/Figure 4A were performed with or without the tail as there is a disconnect between the carton and description. Also, the description of the variants I think varies-- is "non-C-tail phosphorylated Akt" the same as" full-length AKT mutations in the absence of C-tail modification"?

– Earlier work from this lab suggests that the linker between the PH and kinase domain influences inhibition. As this region was missing in experiments using the PH domain in trans, could the authors speculate on how the lack of the linker would influence those results?

– Arguments that the PH domain of R86A retains the same fold as WT but is more stable/less dynamic are indirectly and weakly supported. While this claim is not critical to the later arguments, the data presented are not compelling. The authors claim that the WT PH domain was less stable b/c it degraded during the NMR experiments but do not show data supporting that claim. They then conclude the structures are only slightly changed between WT and R86A based on the ~60% of residues they can assign. If the WT PH domain is degrading during the course of the experiment, how legitimate are the assignments from this experiment and how do they know they are looking at a degraded protein? The analysis of the B-factor of the crystal structure also does not provide much insight into dynamics, especially without any comparison to the b-factor distribution of the WT PH domain – A quick look in pymol reveals a similar pattern of B-facto distributions for the WT PH Domain. Further in 7MYX the loop 42-49 is disordered but is well resolved in the WT structure (1UNP). Since Chu 2020 *eLife* says this loop is important for binding C-tail and PIP3 – what significance does the additional disorder mean? This disordered loop also implies the statement that the two structures of PH domains are in "high concordance" and "identical" is an overstatement (pg7).

– Can the authors speculate at the precise step that the PH domain is inhibiting AKT? The Α Fold 2 structure looks to have the activation loop in the out (ie activated state) even though there is no phosphorylation on the activation loop for AlphaFold2 (relatedly a description of the activation state of the AlphaFold2 structure may be useful). If the Y18:F309 interaction only happens when the activation loop is phosphorylated, is the autoinhibition by the PH domain only at the step of C-tail phosphorylation and does that interaction require an activation loop phosphorylation? Does the PH domain interaction happen equally as strongly with a variant in which the AL is unphosphorylated? In cells Y18A AKT lacks C-tail phosphorylation. As upstream kinases are responsible for this C-tail phosphorylation, how does the PH domain:Kinase domain interaction block C-tail phosphorylation?

*Reviewer #3 (Recommendations for the authors):*

1. Validation data that should be included in the intact MS data for all Akt constructs characterised in this manuscript, as well as the gel filtration traces, showing monodispersity for all constructs. The authors mention in the conclusion that they see phosphorylation of the turn motif, but no data is included here. Previous work has shown that the ligation strategy employed here leads to protein lacking turn motif phosphorylation (although this may be due to the truncated linker), however, if this is the case here this may cause the protein to be less stable, however intact MS traces will also validate that there are no other differences between the generated proteins.

2. The authors should alter their results and discussion to more clearly describe some of the uncertainty of their analysis of the alphafold model. At a minimum, the inclusion of the predicted aligned error, highlighting some of the uncertainty of the PH-kinase interface needs to be included. In addition, the results and Discussion section need to be rewritten to describe this point. The 309L mutant does not seem definitive, and molecular conclusions should be subdued.

---

## [Author Response]

Essential revisions:There are a number of reviewer suggestions to further strengthen this manuscript (see specific reviewer comments below). In particular, two important issues are identified that the reviewers request be addressed in a revised manuscript:1) Questions are raised about the validity and predictive value of the AlphaFold model. Please see specific reviewer comments below and address them accordingly.

We share the general concern that the confidence level in the AlphaFold predicted model of the autoinhibited Akt state is modest. However, as we mention, the experimentally determined crystal structures also have their limitations. We by no means believe that the AlphaFold model should be accepted uncritically but we are of the opinion that it is a publicly available resource for the community to inspect and analyze. As far as what the key templates were that drove the AlphaFold model, we are of course unsure, but there appear to be elements of the active kinase domain structure in the absence of the PH domain which date back to the early 2000s. However, for the phospho-Thr308 state, the form used in our studies, we think that an active kinase domain conformation is reasonable. In any event, we used the AlphaFold structure for hypothesis generation, namely to examine a possible unexpected interaction between Tyr18 and Phe309. The experimental data suggest that this interaction is plausible, although by no means definitively proven. We now summarize these points in the revised manuscript.

2) Further validation of protein samples is requested. Specifically, MS quantification and gel filtration profiles are requested.In addition, the framing of this work creates some confusion. Given that this is submitted as a Research Advance, it is framed in the context of different phosphorylations of the C-tail on AKT. Since the results then focus on side chains in the PH domain the framing, as presented, creates some confusion about the goals at the outset and the eventual findings. The reviewers are confident that the authors can reframe the work in a manner that both explains the connection to earlier findings and clearly steers the reader through the present work.

We now incorporate the requested mass spectrometry and size exclusion chromatography data in the revised manuscript. The mass spec data support that the composition of the semisynthetic Akt proteins prepared by expressed protein ligation are consistent with their proposed structures, taking into account some heterogeneity due to linker phosphorylations as reported previously (A. Salguero et al., ACS Chemical Biology 2022). The SEC profiles show that these Akt proteins appear monomeric and monodisperse.

Reviewer #1 (Recommendations for the authors):The following are provided to improve the quality of this manuscript:1) Figure 3A seems to show multiple peaks for the wt protein possibly indicative of slow exchange (the mutant spectrum appears greatly simplified). The authors do not adequately comment on this or attempt to establish whether the resonances that appear to give rise to multiple peaks are structurally related. Moreover, it is unclear whether the wt sample could be subject to changes in temperature or ZZ-exchange experiments to more carefully analyze the possibility of multiple species in slow exchange. This seems relevant to the idea that the R86A mutation 'stabilizes' the PH domain and in so doing pre-organizes the PH domain for binding to the kinase domain.

We thank the Reviewer for carefully looking at our spectra and proposing this hypothesis. The WT isolated PH domain is not stable and crashes out of solution in a few hours despite of our attempts to find conditions to stabilize the protein. This impedes the possibility of performing long NMR experiments, including relaxation dispersion, CEST and ZZ exchange experiments that report on multiple conformations. Our initial analysis based on the number of signals we observed in the ^1^H-^15^N HSQC led us to the conclusion that there was a single conformation. To illustrate our thought process here, we present Author response image 1. On this figure, the unassigned peaks are indicated by red asterisks and unassigned residues highlighted in red in the primary sequence of our WT PH domain construct. Assuming that the resonances from the Glurich loop and C-terminus residues cluster in the 8.0/122 ppm region that is highly crowded on the spectrum, we count 21 unassigned peaks for 22 unassigned residues. There are four tryptophan residues and in the PH domain and we see four resonances corresponding to the tryptophan side chain atoms at the appropriate spectral region (~130 ppm N and ~10 ppm 1H). This is by no means is a definitive proof of the absence multiple conformations.

However, upon the insightful comment of the reviewer we went back to the data and analyzed the ^1^H-^15^N HSQC spectra recorded at different temperatures. As the reviewer suggested we observed multiple resonances for certain signals at 20 °C, which collapses to a single signal at lower temperatures. When residues corresponding to these signals are plotted on the structure, they cluster in the general vicinity of the R86A site. We further investigated the potential existence of multiple conformations in the R86A mutant and saw no evidence for it from the ^1^H-^15^N HSQC. We have recorded relaxation dispersion experiments on the stable R86A sample and do not observe evidence for exchange on the µs-ms time scale.

**Author response image 1. sa2fig1:** Assigned ^1^H-^15^N HSQC spectrum of WT PH domain at 20 °C. Red asterisks mark unassigned peaks. Top: primary sequence with assigned residues in blue and residues likely corresponding to unassigned peaks (red asterisks) in red.

Thanks to the critical observation of the Reviewer, we now added the following section of text to the manuscript and a new extended figure in support of the statement.

“Though the X-Ray structures of the WT and R86A PH domains are very similar, the NMR chemical shift perturbations at locations distal to the mutation site could be explained by changes in intrinsic dynamics. One observation from the WT PH domain ^1^H-^15^N HSQC spectrum (Figure 3A) supports this hypothesis. A few resonances present a secondary, weaker, signal (Figure 3 —figure supplement 3A). These weak signals could be due to a minor conformation of the same residue or could correspond to an unassigned residue. We investigated these weak resonances as function of temperature and observed that they collapse into a single signal at lower temperatures, supporting the case of multiple exchanging conformations. Furthermore, the residues that display multiple conformations cluster around R86 in a disk-like arrangement reminiscent of the CSPs between WT and R86A (Figure 3 —figure supplement 3B). Taken together with its lower stability and melting temperature, this collectively suggests the possibility of multiple exchanging conformations in WT PH domain. However, the limited stability of the sample impedes further investigation by NMR thus we cannot fully confirm the presense of multiple conformations.

It should be noted that the ^1^H-^15^N HSQC spectrum of the R86A PH domain does not contain these weak, secondary, signals (Figure 3 —figure supplement 3D) which suggest that R86A mutation removes some of the WT dynamics. The improved stability of R86A PH domain allowed us to conduct relaxation-dispersion experiments that further confirmed the absence of dynamics on the µs-ms timescale (Figure 3 —figure supplement 3E). We went on and measured the R_1_, R_2_, Eta_xy_ relaxation rates and heteronuclear Overheauser effect (hetNOE) for the R86A mutant (Figure 3 —figure supplement 4). These relaxation rates show high rigidity (most hetNOE values > 0.65 and calculated correlation time of 8.7 ns, expected for a globular, rigid protein of 14 kDa). Together with the observed high stability and melting temperature of R86A, this proves the absence of multiple exchanging conformations in the R86A PH domain.

Thus there is a strong possibility that the WT PH domain, in contrast to the R86A mutant, is significantly more dynamic and that this could be the reason for the significant chemical shift perturbations. These results may suggest that the PH domain conformation of the isolated R86A PH domain is preorganized before binding to the kinase domain to effectively autoinhibit Akt.”

2) The notion that the R86A mutation stabilizes the PH domain seems to be based on the longevity of the protein sample. Could a more direct measure of protein stability that does not require the long acquisition times of the NMR relaxation experiments (for example, melting temperature) be used to support this idea?

We are grateful to the Reviewer for this suggestion. We have performed thermal shift differential scanning fluorimetry on WT and R86A PH domains. New Figure 3 —figure supplement 1 shows that the melting temperature of R86A is significantly higher (51°C) than that of WT (42°C).

We added the following sentence to the manuscript:

“Differential scanning fluorimetry revealed that R86A has increased melting temperature (51+/-1 °C) compared to WT PH domain (42+/-1 °C, Figure 3 —figure supplement 1).”

3) In describing the data for Figure 3—figure supplement 3, the authors state that "we observe similar patterns of CSPs and peak broadening indicative of the interaction of the PH domain with the rest of the protein. Though the comparative CSP patterns of the R86A and WT PH domains when anchored to the rest of Akt look similar, there are subtle differences…". I agree with this conclusion but due to the absence of many peaks in the wt spectrum it is hard to pick out the similarities between the WT and R86A mutant as presented – in fact the histograms in Figure 3—figure supplement 3 look quite different. Perhaps the language used to compare the two datasets can be modified to make the point clearly without creating confusion. Importantly, it seems the authors need to clearly show the differences around residues 67-73.

We agree that the phrasing of the CSPs interpretation could create confusion. We sought to clarify this and draw three main conclusions from our data: (i) the affinity of R86A for the kinase domain is higher than WT, resulting in CSPs of higher magnitude, (ii) the R86A CSP pattern is still consistent with our interaction model between PH and kinase domains and (iii) there are however small differences in areas where we did not expect interaction.

The manuscript now reads:

“The average CSPs on the PH domain in the context of FL Akt are higher in the case of the R86A mutant in line with the higher binding affinity. In both cases, we observe similar patterns of CSPs and peak broadening indicative of a similar interaction of the PH domain with the rest of the protein. Specifically, aa 14-22 show high CSPs or peak broadening in both cases and this region is expected to be in the core of the binding interface to the kinase domain and aa81-90 similarly show indications of binding, as expected from our previous model (Chu, Viennet, et al., 2020). Though the comparative CSP patterns of the R86A and WT PH domains when anchored to the rest of Akt look similar, there are subtle differences around residues 67-73 (Figure 3 —figure supplement 5C) which indicates that the R86A mutation might reorganize the interaction surface and/or affinity.”

We made a new Figure 3 —figure supplement 5C that specifically shows the CSPs of Arg67, Arg69 and Asn71.

4) With regard to residues 67-73 it is intriguing that this is where the CSPs seem to differ between wt and R86A. Inspection of the structure shows that R69 in this region is involved in very close interactions with W22 (cation π interaction?). As well, R15 sits on the opposite face of W22 and both R15 and W22 surround the important E17-Y18 segment. Moreover, R69 is surrounded by prolines – could proline isomerization be at play here? Certainly, this is an event that could induce the slow exchange that is suggested in the 1H-15N HSQC spectrum of wt (Figure 3A).

We thank the Reviewer for suggesting this model. It is interesting that these differences in the 67-73 segment arise only in the context of full-length Akt, neither CSPs nor crystal structure show significant differences in this region in isolated PH domains. Unfortunately, the resonances corresponding to Arg15 and Trp22 are broadened beyond detection in full-length Akt spectra, which makes it difficult to assess the model further.

The Pro isomerization inducing multiple conformations in an interesting idea. In response to this question, we looked at the CB shifts of Pro68 and Pro70 in R86A which are 31.9 ppm and 32.8 ppm, respectively, suggesting trans conformations. The poor quality of HNCACB for WT did not allow CB assignments for these Prolines. We do not see clear indications of multiple conformations for the neighboring Arg67, Arg69 or Asn71 in the spectra of isolated WT or R86A PH domains. The residues in the vicinity of the two prolines are broadened beyond detection in the R86A fulllength Akt spectrum preventing further analysis. Hence, our data can neither strongly support nor negate this hypothesis. It should be noted that both these Pro residues are in trans conformations in the crystal structures of the PH domain, both for WT and R86A mutant.

5) In Figure 7A-D. (and Figure 7 supp 1) "PH domain" in blue is the R86A mutant? If so please clarify in the figure and caption – the text indicates the R86A PH domain while the figure is unclear. Assuming Figure 7 is the R86A mutant, was the corresponding data for the wt PH domain also acquired to assess conformational heterogeneity at the Y18 position for wt versus R86A mutant? Such an experiment could help better understand the effects of the R86A mutation on the Y18 sidechain.

We apologize for the lack of clarity in this section and thank the reviewer for pointing it out. Yes, we used the R86A mutant for this experiment to enhance the affinity of the interaction between the PH and kinase domains of Akt. Moreover, the inclusion of R86A mutation in these constructs was helpful for their stability while performing NMR studies. We have now clarified this in the caption for Figure 7A-D. The corresponding data for the WT PH domain would certainly be nice to have. However, the stability and expression level of the WT PH containing 19F-Tyr18 was poor rendering the semisynthesis of full-length 19F-Tyr Akt impractical. Thus, for assessing the effects of the F309L mutation and phosphorylation of T308 on the Y18 sidechain, R86A mutation was included in each of the constructs used in this set of experiments.

Reviewer #2 (Recommendations for the authors):The multi-pronged approach systematically investigates the interaction between PH:Kinase, stability/structure of PH domain, activity of AKT and half-life of activation loop phosphorylation. Methodologically this work is a tour de force with the use of 3-way ligation to generate specific, stoichiometric post-translationally modified variants to study and highlights the limitations of studies using non-physiological variants.This paper is written for an AKT expert and may not successfully connect with the broader eLife audience. The main conclusion of the paper comes quite late and was unanticipated. While such findings are exciting, I found it hard to connect the earlier and later results into a coherent finding.– This paper is written for an expert and as such may not successfully connect with the broader eLife audience. For example, an introduction to the regions of the PH domain responsible for lipid binding and those proposed to bind AKT would help provide context for understanding the early mutagenesis. Perhaps figure 1B could be relabeled to identify these surfaces of the PH domain? Similarly, introducing the models of AKT autoinhibition earlier would help provide a framework for the subsequent experiments.

We thank the Reviewer for the recommendation to highlight regions of the PH domain involved in PIP3 binding and surfaces proposed to interact with the kinase domain. We also agree that introducing some of the details about the prior models for autoinhibition earlier makes sense. We have made these edits in the revised manuscript.

– The paper is framed as a study in the role of different phosphorylations of the C-tail on AKT but ends with a result that is separate from the tail. At the end of the paper, I'm still pondering how the tail could/would influence this new autoinhibitory interaction. Can the authors clarify how the phosphorylation of the AKT tail influences interactions/activities of variants tested and how those states would influence their model? Relatedly, several times throughout the paper I struggled to determine whether the variant used had the tail present and, if so, its phospho-state. I'm still not sure if the experiments in Figure 2C/Figure 4A were performed with or without the tail as there is a disconnect between the carton and description. Also, the description of the variants I think varies-- is "non-C-tail phosphorylated Akt" the same as" full-length AKT mutations in the absence of C-tail modification"?

The Reviewer is correct that we have not resolved the details of how phosphorylation of Ser477/Thr479 can activate. We think that it is possible that these phosphoSer/Thr directly bind Arg86 but have no direct evidence of this. We have added a bit more of this to the Discussion. Regarding which Akt forms were used in a particular experiment, we apologize for the confusion. We have added cartoons like in Chu et al. 2020 to clarify in the revised manuscript (Figure 1 figure supplement 1). Indeed "non-C-tail phosphorylated Akt" is the same as " full-length AKT (mutations) in the absence of C-tail modification" in this manuscript.

– Earlier work from this lab suggests that the linker between the PH and kinase domain influences inhibition. As this region was missing in experiments using the PH domain in trans, could the authors speculate on how the lack of the linker would influence those results?

The Reviewer astutely points out the potential that “in trans” interaction may not recapitulate precisely the intramolecular interaction between kinase and PH domains. As they note, our prior enzymatic studies suggest that linker length and composition influence the PH-kinase intramolecular interaction. Therefore, the intermolecular (in trans) interaction experiments should be interpreted cautiously. Having said that, the intermolecular affinity interactions associated with R86A, Y18A, and E17K correlated fairly well with the full-length semisynthetic enzymatic activities of the Akt forms, suggesting that the “in trans” experiments can capture at least some aspects of the intramolecular recognition.

– Arguments that the PH domain of R86A retains the same fold as WT but is more stable/less dynamic are indirectly and weakly supported. While this claim is not critical to the later arguments, the data presented are not compelling. The authors claim that the WT PH domain was less stable b/c it degraded during the NMR experiments but do not show data supporting that claim. They then conclude the structures are only slightly changed between WT and R86A based on the ~60% of residues they can assign. If the WT PH domain is degrading during the course of the experiment, how legitimate are the assignments from this experiment and how do they know they are looking at a degraded protein? The analysis of the B-factor of the crystal structure also does not provide much insight into dynamics, especially without any comparison to the b-factor distribution of the WT PH domain – A quick look in pymol reveals a similar pattern of B-facto distributions for the WT PH Domain. Further in 7MYX the loop 42-49 is disordered but is well resolved in the WT structure (1UNP). Since Chu 2020 eLife says this loop is important for binding C-tail and PIP3 – what significance does the additional disorder mean? This disordered loop also implies the statement that the two structures of PH domains are in "high concordance" and "identical" is an overstatement (pg7).

Thanks for these comments. Regarding the stability of the R86A PH domain vs the WT PH domain, we have now augmented our prior NMR evidence with differential scanning fluorimetry as discussed above. The R86A has 9 °C increased thermal stability compared to WT (see new Figure 3 —figure supplement 1). We did not observe changes in chemical shifts over the course of our NMR experiments, only reduction of the intensities of the peaks corresponding to the starting conformation. If the reviewer suggests the possibility that the WT PH domain is proteolytically degraded during the course of the experiment, we can conclusively state that this is not the case. In case of proteolytic degradation, we would see additional resonances in a specific part of the ^1^H-^15^N HSQC, which we did not observe. In addition, the resonances will shift as a result of the removal of residues, this was also not observed. Another possibility is that the protein aggregates over time, which we physically observe as precipitation. The aggregated and precipitated protein do not contribute signals in solution NMR. Our time course measurements show no change in peak position rather reduction in intensity, in line with protein aggregation. Therefore, we are confident of the NMR backbone assignments made for the WT and R86A PH domains although concede that the instability of the WT PH domain complicates structural comparisons by NMR. As recommended, we have softened the language in the revised manuscript on the concordance of the two structures given the limitations alluded to.

– Can the authors speculate at the precise step that the PH domain is inhibiting AKT? The Α Fold 2 structure looks to have the activation loop in the out (ie activated state) even though there is no phosphorylation on the activation loop for AlphaFold2 (relatedly a description of the activation state of the AlphaFold2 structure may be useful). If the Y18:F309 interaction only happens when the activation loop is phosphorylated, is the autoinhibition by the PH domain only at the step of C-tail phosphorylation and does that interaction require an activation loop phosphorylation? Does the PH domain interaction happen equally as strongly with a variant in which the AL is unphosphorylated? In cells Y18A AKT lacks C-tail phosphorylation. As upstream kinases are responsible for this C-tail phosphorylation, how does the PH domain:Kinase domain interaction block C-tail phosphorylation?

It would be interesting to know what the affinity of the PH domain is for the nonThr308 phosphorylated kinase domain is, although this has not been measured. We will plan to do this in a future study. We believe that the importance of the PH domain’s inhibition of Akt is most clear when Thr308 is phosphorylated since the non-Thr308 phosphorylated form is intrinsically autoinhibited without the PH domain. We are uncertain about the timing of the Phe309 and Tyr18 interaction and there is insufficient data to make a confident statement about this, although our model is that this Phe309/Tyr18 interaction could occur in the presence of Thr308 phosphorylation. Future experiments can hopefully help address this.

Reviewer #3 (Recommendations for the authors):1. Validation data that should be included in the intact MS data for all Akt constructs characterised in this manuscript, as well as the gel filtration traces, showing monodispersity for all constructs. The authors mention in the conclusion that they see phosphorylation of the turn motif, but no data is included here. Previous work has shown that the ligation strategy employed here leads to protein lacking turn motif phosphorylation (although this may be due to the truncated linker), however, if this is the case here this may cause the protein to be less stable, however intact MS traces will also validate that there are no other differences between the generated proteins.

These mass spectra and gel filtration profiles are now included in the revised manuscript for the key and representative semisynthetic proteins prepared in this study (the majority of those prepared). As reported recently, the mass spectra show that the correct masses are observed for the major expected species (including Thr308 and Thr450 phosphorylation +/-C-terminal phosphorylation) with some heterogeneity indicating additional phosphorylations localized to the linker region. The SEC chromatograms show the position of the monomeric and monodisperse forms of the Akts isolated here.

2. The authors should alter their results and discussion to more clearly describe some of the uncertainty of their analysis of the alphafold model. At a minimum, the inclusion of the predicted aligned error, highlighting some of the uncertainty of the PH-kinase interface needs to be included. In addition, the results and Discussion section need to be rewritten to describe this point. The 309L mutant does not seem definitive, and molecular conclusions should be subdued.

We thank the Reviewer for this suggestion and have attempted to be clearer about the uncertainties around the AlphaFold model and the interpretation of the F309L data. Also we made a new Figure 6 – supplement figure 1 showing the AlphaFold-predicted full-length Akt structure with its confidence score (pLDDT).